# STRIDE: A Tool-Assisted LLM Agent Framework for Strategic and Interactive Decision-Making

## Abstract

Large Language Models (LLMs) like GPT-4 have revolutionized natural language processing, showing remarkable linguistic proficiency and reasoning capabilities. However, their application in strategic multi-agent decision-making environments is hampered by significant limitations including poor mathematical reasoning, difficulty in following instructions, and a tendency to generate incorrect information. These deficiencies hinder their performance in strategic and interactive tasks that demand adherence to nuanced game rules, long-term planning, exploration in unknown environments, and anticipation of opponents' moves. To overcome these obstacles, this paper presents a novel LLM agent framework equipped with memory and specialized tools to enhance their strategic decision-making capabilities. We deploy the tools in a number of economically important environments, in particular bilateral bargaining and multi-agent and dynamic mechanism design. We employ quantitative metrics to assess the framework's performance in various strategic decision-making problems. Our findings show that our enhanced framework significantly improves strategic decision-making capability of LLMs. While we highlight the inherent limitations of current LLMs, we demonstrate the improvements through targeted enhancements, suggesting a promising direction for future developments in LLM applications for interactive environments.

## 1 Introduction

Large language models (LLMs) have demonstrated exceptional proficiency in generating coherent natural language from textual inputs (Bubeck et al., 2023). They display human-like strategic thinking and excel at flexible reasoning with nuanced, context-specific information (Aher et al., 2022; Kwon et al., 2023; Suzgun et al., 2022). These successes have sparked interest in their potential for decision-making in complex environments (Yao et al., 2022; Shen et al., 2024; Wang et al., 2023).

To further integrate LLMs into our society, such as deploying them as fiduciary agents on behalf of individuals or organizations in a competitive environment where human and AI agents coexist, the ability to reason strategically is of vital importance. However, due to their inherent limitations in basic mathematics (Bubeck et al., 2023), instruction following (Jang et al., 2022), and susceptibility to hallucinations (Chen et al., 2023), the following challenges exist: (i) LLMs may fail to accurately interpret game rules and objectives expressed in natural language, e.g., form a well-defined utility function that reflects their preference over possible outcomes (Guo et al., 2023); (ii) LLMs are generally inept at long-horizon planning to maximize their utility, which is essential in scenarios where decisions have extended consequences (Huang et al., 2024); (iii) They exhibit poor capabilities in strategic exploration of unknown environments (Krishnamurthy et al., 2024), which hampers their ability to optimize decisions on unforeseen conditions; (iv) LLMs have limited capacity in anticipating opponents' moves and adapting their strategies accordingly (Park et al., 2024), which is crucial for any competitive interaction. These limitations collectively underscore the challenges in deploying LLMs for nuanced and dynamic strategic reasoning tasks.

This paper seeks to improve the LLM agent's reasoning capabilities in these scenarios with the help of tools and external memory. For evaluation, we choose a collection of decision-making problems that highlight the aforementioned challenges in significant and economically relevant real-world

**Problem Description** The alternating offer bargaining game proceeds over several rounds with a finite deadline $T$, during which the buyer and seller take turns proposing prices. The buyer's value, denoted as $v_b$, is the highest price the buyer is willing to pay, while the seller's value, $v_s$, is the lowest price the seller is willing to accept. Both players' preferences for immediate transactions are captured by their respective discount factors, $\delta_b$ for the buyer and $\delta_s$ for the seller, where $\delta_b, \delta_s \in (0,1)$. If an agreement is reached at a price $p$ in round $t \leq T$, the buyer's utility is $u_b = (v_b - p)\delta_b^{t-1}$, and the seller's utility is $u_s = (p - v_s)\delta_s^{t-1}$. If no agreement is reached by the deadline, the negotiation ends with no transaction, and both players get 0 utility.

Public information $v_b = 10, v_s = 0, \delta_b = 0.7, \delta_s = 0.7, T = 4$

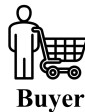

**Buyer**

**Action**: $5
**Message**: I propose a price of $5. This offer ensures an equal utility of 5 for both me and the seller, making it fair and optimal. Accepting this price now grants the seller a better utility than rejecting it and proposing a price in the next time step due to the discount factor.

**Action**: $7.9
**Message**: I reject the proposed price of $5. As a counteroffer, I propose a price of $7.9. If the buyer accepts this price now, the buyer gets a utility of 1.47, which avoids the risk of getting zero utility if we do not reach an agreement by the deadline and the decrease of utility caused by more bargaining rounds. Therefore, accepting my proposed price is optimal for the buyer.

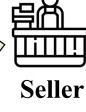

**Seller**

...

Figure 1: This figure illustrates the alternating offer bargaining game with complete information, where the buyer and seller engage in multiple rounds of negotiation over the price of an item using natural language. To make optimal decisions, the agents must extract offer details from the dialogue, predict the opponent's behavior, and evaluate the utilities associated with different decisions.

strategic settings: Bilateral Bargaining Games, where the agents negotiate prices while assessing utility, inferring private values, and predicting opponents' behaviors, addressing Challenges (i), (ii), and (iv) (Rubinstein, 1982; Fudenberg & Tirole, 1991); Markov Decision Process (MDP), where the agent must explore the unknown environment and plan over a long horizon to compute the optimal policy, addressing Challenges (ii) and (iii) (Sutton & Barto, 2018); Dynamic Mechanism Design, a multi-agent extension of MDP where the mechanism designer must anticipate agents' strategic behavior and ensure truthful reporting, covering Challenges (i)-(iv) (Bergemann & Välimäki, 2010; 2019). For each scenario, we provide quantitative metrics to assess the agent's performance. As illustrated by the example of bilateral bargaining problems in Figure 1, these problems necessitate not only the interpretation and response to multiple rounds of dialogue, but also the computation of optimal actions based on the information extracted during interactions. This goes beyond typical LLM tasks, where output is often limited to text generation or language understanding without involving complex strategic reasoning (Yang et al., 2018; Shridhar et al., 2020). Additionally, unlike tasks in which LLMs generate code to solve static math problems (Cobbe et al., 2021; Imani et al., 2023), our agent operates in a dynamic and evolving environment, where it continually interacts with the environment, gathering new knowledge and performing new computations with each interaction.

In light of these unique challenges, we propose a novel LLM agent framework, named STRIDE, which is specifically designed for the multi-step reasoning in ***STR**ategic and **I**nteractive **DE**cision-making* problems. The LLM, which serves as the controller of the whole framework, orchestrates the reasoning process through a sequence of structured *Thought* units. Each *Thought* unit, in addition to typical textual reasoning (Yao et al., 2022), also outlines a series of operations, which are predefined Python functions managing the low-level calculations in various decision-making scenarios. Additionally, an external working memory is integrated to preserve crucial parameters. Therefore, Challenge (i) can be addressed by executing an operation that evaluates the agent's utility in the *Thought* unit. Challenge (ii), which is mainly caused by the information loss in long-context (Liu et al., 2024), can be addressed by extracting and storing important problem parameters and intermediate results in the working memory. Challenges (iii) and (iv) can be addressed through a combination of operations that perform strategic exploration or belief updates. We also explored equipping STRIDE with the ability to synthesize operations on-the-fly when predefined ones are unavailable. Through an extensive evaluation of the selected decision-making problems, we show that, with few in-context examples, STRIDE can make strategic decisions on new problem instances with high success rate. This highlights the transformative potential of integrating LLMs with specialized tools, memory, and control structures to enhance strategic decision-making capabilities.

## 2 RELATED WORK

**Evaluating LLMs' Reasoning in Strategic Environments.** Recent studies have investigated LLMs' capacity for strategic reasoning in settings such as matrix games (e.g., Dictator and Prisoner's Dilemma) (Brookins & DeBacker, 2023; Lorè & Heydari, 2023; Fan et al., 2023; Akata et al., 2023; Guo et al., 2023), focusing on zero-shot prompting to evaluate their ability to act strategically with minimal input. Davidson et al. (2024) and Bianchi et al. (2024) extended this line of work to bargaining games, showing that while LLMs can produce plausible strategies, they often lack consistency and a deep understanding of game dynamics. Few-shot chain-of-thought (CoT) prompting has been proposed to enhance strategic reasoning in matrix and multi-turn bargaining games (Gandhi et al., 2023), but results indicate persistent challenges with complex rules or long horizons. Similarly, Huang et al. (2024) highlighted LLMs' difficulty in generalizing reasoning across diverse game contexts. While these studies advance understanding of LLMs in strategic environments, they treat LLMs as isolated models, relying solely on intrinsic reasoning capabilities without leveraging tools or memory—a gap our work aims to address.

**Optimization in Static and Structured Contexts.** In contrast to dynamic strategic settings, LLMs have been applied to solve static, well-structured optimization problems (Ramamonjison et al., 2023; Tang et al., 2024), where textual problem descriptions are translated into Python code to compute predefined objectives. Some works integrate tools and memory (Xiao et al., 2023; AhmadiTeshnizi et al., 2024; Li et al., 2023), yet these approaches focus on producing single, one-off solutions without iterative adaptation or interaction. Our framework differs fundamentally by addressing dynamic decision-making, requiring LLMs to adapt computations and strategies to evolving environments. Moreover, beyond generating solutions, our approach emphasizes articulating algorithmic reasoning in natural language, as shown in Figure 2. This enhances both interpretability and adaptability—a critical capability absent in static optimization contexts.

**Broader Applications of LLM-based Agents.** LLM-based agents have also found applications in diverse domains, including social simulation to model human behavior (Park et al., 2023; Aher et al., 2023), scientific research for automating experiment design and execution (Boiko et al., 2023), software development using collaborative agents (Qian et al., 2023), and robotics for advanced manipulation and navigation (Ahn et al., 2022). These systems often employ modular architectures with memory (Zhu et al., 2023) and planning modules (Yao et al., 2022), enabling adaptability in dynamic scenarios. However, they prioritize context-aware interactions and flexibility over achieving optimal behavior. While this underscores the broad applicability of LLM agents, our work addresses the distinct challenge of designing agents that not only adapt to dynamic and evolving environments but also consistently achieve optimal performance—bridging the gap between flexibility and precision.

## 3 LLM AGENT FOR STRATEGIC AND INTERACTIVE DECISION MAKING

As shown on the left side of Figure 2, our primary strategy to address the four challenges in Section 1 is to provide the LLM with an operation library, i.e., a set of Python functions taking care of lower-level computation in various decision-making problems and a working memory retaining important parameters. Most importantly, we introduce a reasoning module that acts as the central executive, orchestrating the information flow among components and synthesizing structured *Thought* sequences as illustrated on the left side of Figure 2 to solve complex problems.

### 3.1 REASONING VIA A SEQUENCE OF *Thought* UNITS

To effectively leverage the operation library to make strategic decisions during the interaction, we propose a unique design for the reasoning module, which is empowered by a pretrained LLM like GPT-4 (Achiam et al., 2023) or Claude (Anthropic, 2024), in the STRIDE framework. Take the bargaining game illustrated in Figure 1 as an example. Suppose it is the seller's turn to decide whether to accept the buyer's offer or return a counteroffer. As shown on the top left of Figure 2, the message containing the buyer's offer will be used to prompt the LLM to initiate the reasoning process. Given this message, the LLM generates a *Thought* unit, which is a structured output whose text field provides reasoning about what needs to be done to answer the question and the operations field comprises a sequence of operations that takes care of the necessary computation. As shown on the right side of Figure 2, the first *Thought* unit in the reasoning process decomposes the task of

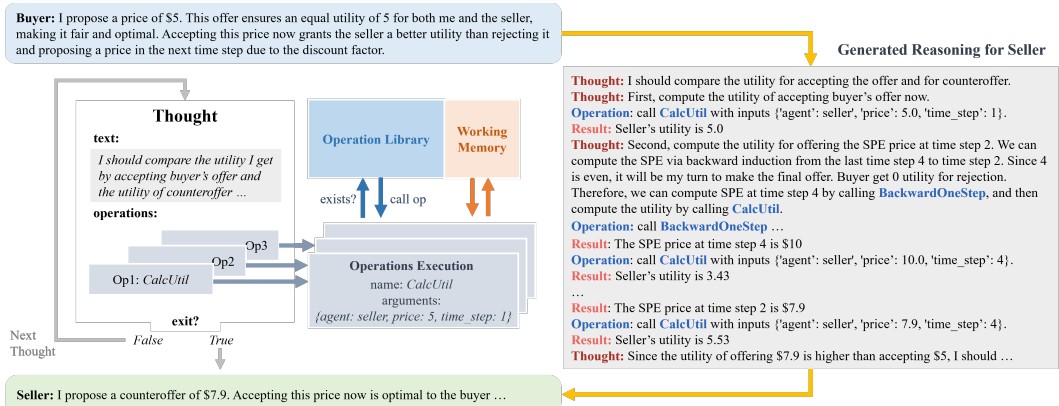

Figure 2: In STRIDE framework, the LLM controls the execution of operations and access to working memory via a sequence of *Thought* units. Each *Thought* unit is a structured data containing three fields: (i) text, which suggests the next step of strategic reasoning and summarize important information; (ii) operations: a list of operations to execute, in order to compute or retrieve information necessary for reasoning; (iii) exit: a boolean value indicating whether the reasoning process is completed. With proper operations and in-context examples, STRIDE can emulate various algorithmic behaviors, e.g. backward induction for bargaining games, to facilitate strategic decision making.

finding the optimal response to the buyer into two subtasks: computing *the utility the seller will get by accepting the offer now*; and computing *the utility the seller will get by making a counteroffer in the next round*, which will be completed with the help of the available operations. In particular, to support decision-making in this problem, the following operations are provided in the library:

- `CalcUtil`: calculate buyer or seller's utility, with the specified price and time step as inputs;
- `BackwardOneStep`: compute the subgame perfect equilibrium (SPE) price using one step of backward induction reasoning based on the opponent's utility if he/she chooses to counteroffer.

As illustrated in Figure 2, *CalcUtil* can be used to compute the seller's utility for accepting the buyer's offer now. Similarly, the seller's utility for making a counteroffer in the next round (assuming rational buyer behavior) can be computed by first calling *BackwardOneStep* repetitively to get the SPE price for the next round and then calling *CalcUtil* to get the corresponding utility. After specifying the required operations in the *Thought* unit, the LLM then extracts the relevant input variables from the context, saves them in the working memory, and then executes the operations to obtain the results. In the end, the decision of whether to accept the offer or make an counteroffer can be decided by comparing the utilities of the two options as shown at the bottom of Figure 2.

**Error Handling and Reflection.** Some additional measures are used to improve the robustness of the reasoning process. Before executing the selected operations, the *Thought* unit undergoes a validation process based on predefined rules to ensure its integrity. For example, a common rule applied in our experiments is the mutual exclusivity of the exit condition and the presence of operations: the *Thought* unit must not simultaneously specify an exit as true while containing non-empty operations, as this often indicates a premature termination of the reasoning process. If this conflict occurs, the system will generate an appropriate error message and prompt the LLM to revise the *Thought* unit. This mechanism ensures that operations proceed only with validated and logically consistent instructions. Enhanced applications of this functionality involve utilizing an additional LLM to verify whether the newly generated *Thought* unit adheres to the reasoning logic and language style presented in the in-context examples. This step can improve consistency and prevent hallucinations. With the *Thought* unit validated, the selected operations will be executed in the specified order. The outcomes of these operations are then utilized to generate the subsequent *Thought* unit. This process continues until *Exit* is set to be true, signaling the completion of the reasoning process.

**Working Memory.** As mentioned in Section 1, for long-horizon planning, LLMs may forget or neglect important information mentioned early in the context. Moreover, an accurate description of the problem instance sometimes require parameters of high dimensions, e.g., transition matrices of MDP. In this case, storing these parameters in the context history is costly and prone to error.

Therefore, STRIDE is equipped with a working memory, i.e., a Python dictionary, that stores the parameters extracted from the context, as well as intermediate results computed by the operations.

**Operation Library.** What sets our work apart from methods like ReAct (Yao et al., 2022) and Reflexion (Shinn et al., 2024) is the sophisticated integration of the operations by the *Thought* sequence to execute complex calculations during text-based reasoning and interactions. For instance, these operations can calculate the utility of the agent based on the outcomes of a game or update the belief about uncertainty on the environment or the other agents. A combination of such operations allows STRIDE to implement various algorithmic behaviors such as dynamic programming to solve MDPs and Bargaining Games, facilitating a deeper and more precise decision-making process. They also let STRIDE scale to complex problems by abstracting detailed computations. This scalability is crucial in handling larger and more challenging scenarios. In addition to strategic decision-making, STRIDE offers a flexible framework that can be extended to a diverse array of problem domains, where algorithmic behavior of LLMs is critical. To tailor STRIDE to other domains, it suffices to construct domain-specific operations and in-context examples to emulate other algorithms using these operations. As we will see in the sequel, STRIDE can be applied to MDP, dynamic mechanism design, two-player bargaining games, Tic-Tac-Toe, Connect-N, etc.

**Generation of In-Context Examples** In-context examples teach the LLM to combine operations in a structured way to emulate reference algorithms—standard methods for solving decision-making problems, such as backward induction for computing SPE in bargaining games and value iteration for computing optimal Q-values in MDPs. These algorithms serve as benchmarks for the target behaviors we aim to replicate. To generate effective in-context examples:

- **Implement Reference Algorithms:** Each reference algorithm is implemented using operations provided in the library, such as `CalcUtil` and `BackwardOneStep` in Figure 2. These modular implementations ensure that the algorithms are expressed in terms of the same reusable operations the LLM will employ, making the connection between the algorithm and the operations explicit.
- **Add Explanatory Comments:** Each algorithm step is annotated with natural language comments explaining its purpose and logic. For well-known algorithms, such as value iteration for MDPs, these explanations can often be generated automatically by LLMs. However, for less popular or novel solutions, manual descriptions are necessary to ensure the logic is accurately conveyed.
- **Generate Worked Examples:** The augmented algorithms are run on sampled problem instances, producing sequences of operation calls, intermediate results, and explanatory comments to demonstrate how to solve the problem using the operations.

The definition of reference algorithms and the added comments are given in Appendix B for clarity and reproducibility. This process equips the LLM with clear, structured demonstrations, making it reason and compute effectively while aligning its behavior with the logic of the reference algorithms.

## 3.2 Operation Synthesis

As outlined in Section 1, our goal is to integrate LLMs into real-world applications where they can act as fiduciary agents in competitive environments. To ensure the robustness and reliability needed for these tasks, STRIDE is primarily built to use *pre-defined, validated operations*. This approach allows us to optimize complex computations and minimizes the risk of errors from on-the-fly code generation. By providing reusable and modular components, we ensure reliable performance while allowing the LLM to focus on its strength—synthesizing high-level workflows, as shown in Figure 2, and adapting to various contexts without handling low-level computations. Nevertheless, we have included functionality that allows STRIDE to dynamically generate operations when pre-defined ones are unavailable or when robustness is less critical. These cases could benefit from the LLM's flexibility to create on-demand solutions. While this feature showcases the LLM's adaptability, it remains secondary to our framework's primary focus on the stability provided by pre-defined operations in structured, high-stakes decision-making tasks.

Specifically, we enable this functionality by augmenting the *Thought* structure depicted in Figure 2 with an optional field called *new_operation*. Therefore, as illustrated in Figure 3, apart from choosing to call the operations available in the library, STRIDE can opt to create a new operation via *new_operation*, which specifies the name of the new operation and a description of its functionality. Then an iterative procedure is triggered which alternates between generating Python code to implement this functionality by the LLM and executing the code on a copy of the working memory to get

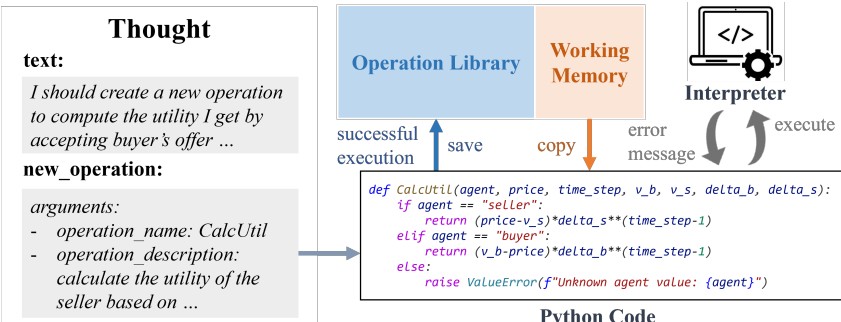

Figure 3: In cases where predefined operations are unavailable or insufficient, e.g. the operation to calculate the seller's utility is missing as shown on the left, STRIDE can dynamically synthesize a new operation in the midst of the reasoning process by defining its name and expected functionality. This information is then used by the LLM to generate Python code, which is executed on a copy of the current working memory for validation. If any syntax or runtime errors occur, they will be used to prompt the LLM to iteratively improve the code. The process repeats until successful execution, after which the generated code is saved as a new operation.

feedback from the code interpreter. Upon successful execution, the generated Python code, together with its description, will be saved as a new operation in the library for future use.

## 4 EXPERIMENTS

For each decision-making problem mentioned in Section 1, we first construct the relevant operations, so that STRIDE is able to emulate the reference algorithm when solving each problem. Descriptions of the selected reference algorithms, the constructed operations, as well as the procedure to generate the in-context examples, can be found in Appendix B. To evaluate whether STRIDE can reliably solve new problem instances given provided in-context examples, we repeat experiments on randomly sampled instances and report the averaged results. We include the following baselines for comparison: (i) *zero-shot Chain-of-Thought (CoT)*, (ii) *zero-shot CoT with code interpreter*, and (iii) *few-shot CoT with code interpreter*, where the latter two can utilize the coding capability of LLMs (through OpenAI Assistants API) to write and execute programs to solve the decision-making problems. Compared with (ii), (iii) is additionally provided with example implementation of the reference algorithm for each problem. Prompts used in all the experiments are given in Appendix C. We also conducted additional experiments on other problem setups like Tic-Tac-Toe and Connect-N games to further demonstrate the generality of STRIDE. Details about these experiments are given in Appendix D.

### 4.1 MARKOV DECISION PROCESSES

We first evaluate STRIDE and the baselines (GPT-3.5-Turbo-0125 with temperature set to $0$ is used for all agents) on MDPs with both known model, where the transition function $P$ and reward function $R$ are given to the agent at the beginning, and unknown model, where the agent needs to estimate $P$ and $R$ during online interactions. In the following paragraphs, we first provide a formal definition of the objective of the agent under each setting and then discuss the experiment setup and results.

**Agent's Objective in MDP with Known Model.** We consider a finite-horizon MDP, where the agent interacts with the environment for some fixed $H$ steps. At each step $h = 1, 2, \ldots, H$, the agent observes the current state $s_h \in \mathcal{S}$, and then chooses action $a_h \in \mathcal{A}$. The environment then produces a reward feedback $R(s_h, a_h)$ to the agent, and then the state transits to $s_{h+1} \sim P(\cdot \,|\, s_h, a_h)$. When the transition function $P$ and reward function $R$ are known to the agent, the objective is to find a policy, denoted as $\pi = \{\pi_h\}_{h=1}^H$ with $\pi_h : \mathcal{S} \to \Delta(\mathcal{A})$ for $h \in [H]$, that maximizes the expected cumulative rewards over $H$ time steps:

$$\max_\pi \mathbb{E}_{\pi, P}\Big[\sum_{h=1}^H R(s_h, a_h)\Big] := V_1^\pi, \tag{1}$$

Table 1: Success rate in taking the optimal action (20 runs).

| H | S | A | zero-shot CoT | zero-shot CoT w/ code | few-shot CoT w/ code | STRIDE(VI) | STRIDE(MCTS) |
|---|---|---|---|---|---|---|---|
| 5 | 3 | 3 | 0.58 | 0.74 | 0.70 | **0.98** | 0.87 |
| 10 | 3 | 3 | 0.62 | 0.75 | 0.69 | **0.87** | 0.80 |
| 5 | 10 | 10 | 0.24 | 0.48 | 0.60 | **0.96** | 0.82 |
| 10 | 10 | 10 | 0.21 | 0.50 | 0.68 | **0.82** | 0.74 |

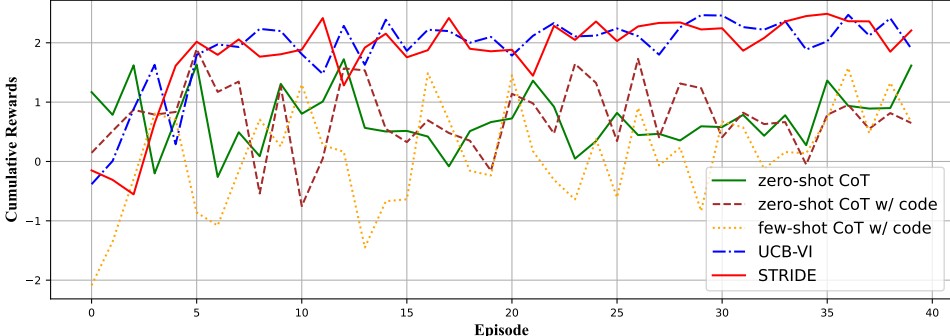

Figure 4: Comparison of cumulative rewards over episode. We observe that both STRIDE and UCB-VI exhibit rapid increases in their cumulative rewards, converging by approximately the 10-th episode. This indicates that STRIDE can effectively explore the environment by emulating UCB-VI in its reasoning. In contrast, the cumulative rewards of other baseline methods display ongoing fluctuations throughout the episodes, showing poor exploration ability in uncertain environments.

where the expectation is with respect to the randomness in state transitions and the stochasticity of $\pi$. There are numerous ways to compute the optimal policy. Here we consider two reference algorithms, i.e., value iteration (VI) and MCTS for STRIDE. Let's denote the optimal Q value function as $Q_h^\star(s, a)$ for $h \in [H]$. Then for any state $s_h$ encountered by the agent at step $h \in [H]$, we check whether the action $a_h$ taken by the agent satisfies $a_h = \arg\max_{a \in \mathcal{A}} Q_h^\star(s, a)$, and report the average success rate in the following experiment.

**Experiment Setup and Results.** We evaluate on MDPs with horizon length $H \in \{5, 10\}$, number of states $|\mathcal{S}| \in \{3, 10\}$, and number of actions $|\mathcal{A}| \in \{3, 10\}$. For each configuration, we repeat the experiment for 20 times on randomly generated instances, by sampling dense tensors of size $\mathbb{R}^{S \times A \times S}$ and $\mathbb{R}^{S \times A}$ as the transition function and reward function, respectively. The average success rates are reported in Table 1. For STRIDE, we only provide it with a single in-context example that solves a MDP instance with $H = 5, S = 5, A = 5$. We can see that STRIDE, either emulating VI or MCTS, outperforms the baselines in taking the optimal actions on the random MDP instances.

**Agent's Objective in MDP with Unknown Model.** In this setting, $P$ and $R$ are unknown to the agent, but the agent is allowed to repetitively interact with the same MDP instance for $K$ episodes to explore and update its belief about $P$ and $R$ using the observed feedback. The agent's objective is to choose a sequence of policies $\pi^1, \pi^2, \dots, \pi^K$ to minimize the cumulative regret:

$$\min_{\pi^1, \pi^2, \dots, \pi^K} \sum_{k=1}^K \left( V_1^{\pi^\star} - V_1^{\pi^k} \right). \tag{2}$$

In addition to the challenge of long-horizon planning exemplified by equation 1, equation 2 also requires addressing the exploration-exploitation dilemma. Specifically, the agent needs to strategically balance between exploring unfamiliar state-action pairs to learn $P$ and $R$, and exploiting the current knowledge about $P$ and $R$ to obtain more rewards. A classic solution to this problem is *UCB-VI* (Azar et al., 2017), which is used as the reference algorithm for STRIDE. To help the baselines work with long context history ($K \times H$ interactions in total), an external summary of the past episodes is added in their prompts at the beginning of each episode, similar to Krishnamurthy et al. (2024).

**Experiment Setup and Results.** In addition to STRIDE and the aforementioned baselines, we also include *UCB-VI* algorithm in the experiments, which serves as a reference. We evaluate on 10 randomly generated MDP instances with $H = 5$, $|\mathcal{S}| = 3$, and $|\mathcal{A}| = 3$, with the agents repetitively playing each instance for a total number of $K = 40$ episodes, and average the results over the 10

Table 2: Success rate in computing the VCG mechanism (10 runs).

| N | zero-shot CoT | zero-shot CoT w/ code | few-shot CoT w/ code | STRIDE |
|---|---|---|---|---|
| 2 | 0.69 | 0.63 | 0.70 | **0.89** |
| 4 | 0.57 | 0.63 | 0.54 | **0.90** |
| 6 | 0.49 | 0.45 | 0.44 | **0.86** |

instances. In Figure 4, we report how the cumulative rewards collected in each episode change as the number of episodes experienced by the agent increases. STRIDE reliably implements the behavior of *UCB-VI* algorithm using the provided operations, and thus converges to the optimal policy at a similar rate as *UCB-VI*. In comparison, the baselines, though given additional summarization of history, fail to find the optimal policy as they cannot efficiently explore the environment.

## 4.2 DYNAMIC MECHANISM DESIGN

Section 4.1 presents the challenges of long-horizon planning and strategic exploration in MDP, which only involves a single agent. Here we further evaluate STRIDE (GPT-4o-2024-05-13 with temperature set to $0$) on dynamic Vickrey-Clarke-Groves (VCG) mechanism design problem (Bergemann & Välimäki, 2019), a multi-agent generalization of MDP, which further necessitates the agent's ability to anticipate other agents' behaviors and plan accordingly.

**Agent's Objective in Dynamic Mechanism Design.** Consider a mechanism designer and a set of $N$ agents. The mechanism designer needs to elicit the reward functions $\{\widetilde{R}_i\}_{i=1}^N$ from the $N$ agents, with each $\widetilde{R}_i : \mathcal{S} \times \mathcal{A} \to \mathbb{R}$, and the agents can be untruthful. Based on reported reward functions, the designer chooses a policy $\pi : \mathcal{S} \to \Delta(\mathcal{A})$. At each step $h \in [H]$, the designer takes action $a_h \sim \pi(s_h)$, e,g., the allocation of some scarce resource among $I$ agents, and each agent $i \in [N]$ receives reward $R_i(s_h, a_h)$, i.e., agent $i$'s valuation for $a_h$ at state $s_h$. After $H$ steps of interactions, the designer needs to charge each agent $i$ some price $p_i \in \mathbb{R}$. The objective of each agent $i$ is to maximize its utility $u_i(\widetilde{R}_i) = V^\pi(P, R_i) - p_i$ by strategically reporting the reward function $\widetilde{R}_i$. The objective of the designer is to maximize the expected cumulative sum of rewards, by strategically choosing the policy and pricing rule. This can be formulated as the following optimization problem

$$\pi^\star, \{p_i^\star\}_{i\in[N]} := \max_{\pi, \{p_i\}_{i\in[N]}} V^\pi(P, \textstyle\sum_{i=1}^n \widetilde{R}_i) \tag{3}$$
$$s.t. \quad u_i(R_i) \geq u_i(R_i'), \forall R_i', i$$

where the constraint guarantees the incentive compatibility of all agents. The success rate for the experiments on this problem is computed by considering: (i) whether the chosen action $a_h$ satisfies $a_h = \pi_h^\star(s_h)$ for $h \in [H]$; and (ii) whether the charged price $p_i$ satisfies $|p_i - p_i^\star| \leq 0.01$.

**Experiment Setup and Results.** We evaluate on problem instances with horizon $H = 5$, number of states $|\mathcal{S}| = 3$, number of actions $|\mathcal{A}| = 3$, and number of agents $N \in \{2, 4, 6\}$. For each configuration, we repeat the experiment 10 times on randomly generated instances, by sampling dense tensors of size $\mathbb{R}^{S \times A \times S}$ and $\mathbb{R}^{N \times S \times A}$ as the transition function and reward functions for $N$ agents, respectively. The average success rate are reported in Table 2. We observe that the baselines, despite being capable of computing the optimal action most of the times, cannot generalize the same value iteration procedure to compute the VCG price correctly. In comparison, STRIDE can reliably compute the VCG price on most problem instances, which leads to its higher success rate.

## 4.3 BARGAINING GAMES

We further evaluate STRIDE and the baselines (GPT-4o-2024-05-13 with the temperature set to $0$) on bargaining games, in which a buyer and a seller engage in repeated negotiation for a finite number of steps. In order to maximize their utility, both the buyer and the seller need to predict the response of their opponent over long-horizon, based on the potentially incomplete information they have.

**Alternating Offer Bargaining under Complete Information.** We first consider the elementary yet seminal setting in which a buyer and a seller engage in a $T$-step bargaining process (with $T < \infty$) over price $p$ of the good. Specifically, at time step $t = 1$, the buyer offers a price to the seller and the game ends if the seller accepts the offer. Otherwise, the game continues to the next time step $t = 2$,

Table 3: Success rate in reaching SPE of single-issue bargaining (10 runs).

| $T$ | zero-shot CoT | zero-shot CoT w/ code | few-shot CoT w/ code | STRIDE |
|---|---|---|---|---|
| 3 | 0.50 | 0.35 | 0.50 | **0.79** |
| 6 | 0.50 | 0.27 | 0.46 | **0.91** |
| 9 | 0.34 | 0.18 | 0.27 | **0.74** |

Table 4: Outcomes of STRIDE and zero-shot CoT bargaining with each other.

| | STRIDE buyer vs zero-shot CoT seller | | zero-shot CoT buyer vs STRIDE seller | |
|---|---|---|---|---|
| $T$ | avg SPE price | avg sale price | avg SPE price | avg sale price |
| 3 | 0.13 | 0.13 | 0.22 | 0.43 |
| 6 | 0.57 | 0.56 | 0.65 | 0.70 |
| 9 | 0.28 | 0.27 | 0.49 | 0.70 |

Table 5: Success rate in reaching SE of single-issue bargaining with one-sided uncertainty (10 runs).

| $T$ | zero-shot CoT | zero-shot CoT w/ code | few-shot CoT w/ code | STRIDE |
|---|---|---|---|---|
| 3 | 0.47 | 0.29 | 0.38 | **0.79** |
| 6 | 0.44 | 0.32 | 0.30 | **0.75** |
| 9 | 0.49 | 0.38 | 0.23 | **0.69** |

where the seller makes a counteroffer. They repeat this process until the deadline $T$ is reached. Assuming the buyer's value for the item is 1 and the seller's cost is 0, then the utility function of the buyer, denoted as $u_b$, and that of the seller, denoted as $u_s$, for some price $p$ at time step $t$ are

$$u_b(p,t) = (1-p) \cdot \delta_b^{t-1}, \text{if } t \leq T, \text{ and } 0 \text{ otherwise};$$
$$u_s(p,t) = (p-0) \cdot \delta_s^{t-1}, \text{if } t \leq T, \text{ and } 0 \text{ otherwise.}$$
(4)

respectively, with $\delta_b, \delta_s \in [0, 1]$ being the discount factor of their utilities over time. Note that in this setting, the buyer's value 1, the seller's cost 0, and the values of $\delta_b, \delta_s$ and $T$ are public information. The optimal decision for either agent, assuming his/her opponent is also acting optimally, i.e., being rational, is to play the Subgame Perfect Equilibrium (SPE) strategy, which, in this setting, is unique and can be computed using backward induction (Fudenberg & Tirole, 1991). Description of this reference algorithm and the operations constructed for STRIDE is given in Appendix B. To evaluate whether STRIDE and the baselines can make optimal decisions, we let buyer and seller empowered by the same method to bargain with each other, and report the success rates in reaching SPE.

**Experiment Setup and Results.** We evaluate on bargaining games with deadline $T \in \{3, 6, 9\}$. In each case, we repeat the experiments on 10 randomly generated instances, by sampling discount factors $\delta_b, \delta_s \in \mathcal{U}(0.5, 1.0)$. The average success rates are reported in Table 3. We can see that, none of the baseline methods attains success rate higher than 0.5, which is because when it is their turn to offer, they cannot offer a price close to SPE, though being explicitly instructed in the prompt to assume rational opponent behavior when making decisions. It is worth noting that the existence of the code interpreter did not provide any advantage this time compared with the results for MDP. Though the LLM did attempt to implement the backward induction algorithms to solve SPE, they failed to get everything right and produce the correct results. We hypothesize that this distinction is due to the insufficiency of training data related to the implementation of backward induction algorithms for bargaining, especially compared with the algorithms for MDP.

Moreover, to further illustrate the advantage of being able to strategically reason about the decisions in bargaining, we pit STRIDE against zero-shot CoT, the best-performing baseline in Table 3. The results (averaged over 10 randomly generated instances) are summarized in Table 4. We can see that, by emulating the reference algorithm, STRIDE guarantees an outcome that is no worse than SPE regardless of the role it plays. As mentioned in the previous paragraph, the baseline cannot accurately compute SPE price, and thus, when it serves as the buyer who needs to make the initial offer, often ends up with a sale price that is higher than SPE price, which shows its sub-optimality.

**Seller Making Offers under Uncertainty of Buyer's Value.** Now we consider a more challenging scenario where the buyer's value, denoted as $b \in [0, 1]$, is privately known to himself, and thus the seller needs to update the belief about $b$ based on the observed responses, i.e., buyer's rejection of

Table 6: Success rate in taking the optimal action (10 runs).

| H | S | A | few-shot CoT w/ code | STRIDE | STRIDE-SynOp |
|---|---|---|---|---|---|
| 5 | 3 | 3 | 0.88 | **0.94** | **0.94** |
| 10 | 3 | 3 | 0.86 | **0.87** | 0.84 |
| 5 | 10 | 10 | 0.80 | **0.90** | 0.86 |
| 10 | 10 | 10 | 0.79 | **0.82** | 0.81 |

seller's offers. The seller's cost (still assumed to be 0) and the prior distribution of $b$, represented as a cumulative distribution function $F(v)$, are public information. $F(\cdot)$ is supported on $[0, 1]$ and we assume $F(v) = v$, i.e., a uniform distribution. In each step $t = 1, 2, \ldots, T$, the seller offers a price and the buyer responds by acceptance or rejection. Similar to equation 4, the utility functions are

$$u_b(p, t) = (b - p) \cdot \delta_b^{t-1}, \text{if } t \leq T, \text{ and } 0 \text{ otherwise},$$
$$u_s(p, t) = p \cdot \delta_s^{t-1}, \text{if } t \leq T, \text{ and } 0 \text{ otherwise}. \tag{5}$$

Different from the complete information setting where we evaluate the agents using the unique SPE, here we consider sequential equilibrium (SE) due to the uncertainty on the buyer's value. Fortunately, in the particular setting described above, the SE is still unique (Cramton, 1984), and thus we can similarly evaluate the agents using the success rates of reaching SE. To compute the SE, we propose a reference algorithm for STRIDE that combines bisection search and backward induction and construct the specialized tools. More details are given in Appendix B.

**Experiment Setup and Results.** We evaluate the agents on problems with deadline $T \in \{3, 6, 9\}$. In each case, we repeat the experiments on 10 randomly generated instances, by sampling discount factors $\delta_b, \delta_s \in \mathcal{U}(0.5, 1.0)$ and buyer's value $b \in \mathcal{U}(0.1, 0.9)$. The average success rates are reported in Table 5. Again, we observe that STRIDE outperforms the baseline methods, as it is able to compute the SE by emulating the reference algorithm we designed.

### 4.4 EVALUATION OF OPERATION SYNTHESIS

To assess STRIDE's ability to dynamically synthesize operations, we designed an experiment in which STRIDE is evaluated on the same MDP environments as in Section 4.1, but with the pre-defined operations deliberately withheld. Therefore, STRIDE needs to create the necessary operations using the procedure discussed in Section 3.2. We denote the resulting method as STRIDE-SynOp. We use GPT-4o-2024-05-13 with a temperature setting of 0 for generating both the *Thought* sequence and the Python code, as depicted in Figure 3. For the iterative code generation process, we set the maximum number of retries to 6, meaning the run is considered a failure if the LLM cannot produce executable code within six attempts for any operation. In Table 6, we compare the average success rate of taking optimal actions achieved by STRIDE-SynOp, STRIDE, and few-shot CoT with code (all powered by GPT-4o-2024-05-13). Despite having to synthesize the operations on the fly, STRIDE-SynOp demonstrates a success rate that remains relatively close to STRIDE across all tested scenarios.

## 5 CONCLUSION

This paper presented the STRIDE framework, enhancing LLMs' strategic decision-making capabilities. Through integrating a structured *Thought* process with external working memory and operations, STRIDE enables LLMs to overcome significant limitations such as strategic exploration and dynamic opponent interaction. Our evaluations across diverse decision-making scenarios validate STRIDE's effectiveness, suggesting its potential as a robust tool for strategic thinking in complex environments. For further development of the STRIDE framework, we propose the following research avenues. (i) Currently, STRIDE utilize specially designed Python functions as tools to facilitate the formation of strategies and the choice of actions by the agents in bilateral bargaining, an interesting direction is to replace it with models trained using data collected during interactions. (ii) Fine-tuning on the *Thought* Sequence: To further enhance LLM's understanding and execution of the *Thought* sequence as well as the associated operations, we can fine-tune the model on the Thought sequences resulting in successful decisions.

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

---

**Algorithm 1** Value Iteration for MDPs with Known Model.

---

1: **Initialize** $V_{H+1}(s) = 0, \forall s \in \mathcal{S}$
2: ▷ **Question: Compute the Optimal Policy.**
3: **for** step $h = H, H - 1, \cdots, 1$ **do**
4:     ▷ Thought: Now we can continue to compute the Q-values for the current step $h$.
5:     ▷ Operation: call `UpdateQbyR` with inputs {time_step: h}
6:     ▷ Operation: call `UpdateQbyPV` with inputs {time_step: h}
7:     ▷ Operation: call `UpdateVbyQ` with inputs {time_step: h}
8:     **for** each state $s \in \mathcal{S}$ **do**
9:         **for** each action $a \in \mathcal{A}$ **do**
10:             $Q_h(s, a) = R(s, a) + \sum_{s' \in \mathcal{S}} P(s'|s, a) V_{h+1}(s')$
11:         $V_h(s) = \max_{a \in \mathcal{A}} Q_h(s, a)$
12: ▷ Thought: I have finished value iteration. Now exit reasoning.
13: **for** step $h = 1, 2, \cdots, H$ **do**
14:     Observe state $s_h$
15:     ▷ **Question: Which action I should take?**
16:     ▷ Thought: I should choose the action that maximizes the computed Q values.
17:     ▷ Operation: call `GetQ` with inputs {time_step: h, cur_state: $s_h$}
18:     ▷ Operation: call `GetArgMax` with inputs {q_vals: [. . .]}
19:     ▷ Exit: I should choose Action $a_h$ as it maximizes the Q values. Now exit reasoning.
20:     Take action $a_h = \arg\max_{a \in \mathcal{A}} Q_h(s_h, a)$
21:     Observe reward $r(s_h, a_h) = R(s_h, a_h) + \epsilon$ and state transits to $s_{h+1}$

---

# A  ADDITIONAL DISCUSSION OF RELATED WORK

**LLM-Enhanced Reinforcement Learning Algorithms.** The works mentioned in the previous two paragraphs, as well as the STRIDE framework proposed in this paper, utilize LLMs as the decision maker, that is, LLMs are fed prompts containing the current state of the environment, and they generate actions based on this input. The reasoning process that produces the recommendation, regardless of whether it follows certain algorithmic behavior as STRIDE, happens in the language space. Another distinct line of research integrates LLMs into traditional reinforcement learning algorithms to leverage the common sense knowledge that LLMs acquire during pretraining (Hao et al., 2023; Liu et al., 2023; Zhou et al., 2023; Zhao et al., 2024). In this way, the reasoning process is hard-coded in programming language like Python, which defines how different components interact with each other. Currently, the most prevalent approach in this domain is the integration of LLMs into Monte Carlo tree search (MCTS) algorithms, where they typically serve as tree traversal policy (Zhao et al., 2024), action pruner (Liu et al., 2023), world model (Hao et al., 2023), and evaluation function (Liu et al., 2023). In comparison, our approach is more flexible in the sense that we can repurpose the reasoning process of STRIDE to emulate different algorithmic behaviors. In particular, as demonstrated in our experiments, apart from the model-based algorithms like UCB-VI, we can also make STRIDE reason as tree-search algorithms like MCTS and Minimax.

# B  REFERENCE ALGORITHMS FOR STRIDE

As discussed in Section 3, the main strength of STRIDE lies in its capability of emulating various algorithmic behaviors in its *Thought* process to solve decision-making problems that are challenging to LLMs. In this section, we provide the descriptions of the reference algorithms that STRIDE emulates when solving the problems in Section 4.

## B.1  VI, MCTS AND UCB-VI ALGORITHMS FOR MDPS

For MDP with known model, the reference algorithms selected for STRIDE are VI and MCTS. For MDP with unknown model, the reference algorithm is Upper Confidence Bound Value Iteration (UCB-VI). Here we provide description of these three algorithms in Algorithm 1, Algorithm 2, and Algorithm 3, as well as some simplified comments (e.g., results returned by the operations are

omitted for simplicity) showing how we augment the algorithm to generate the in-context examples for the LLM.

**Operations.** The following operations are provided to the LLM to help it emulate VI and UCB-VI:

- `UpdateQbyR`: add reward $R(s, a)$ to $Q_h(s, a)$ for all $(s, a)$ pairs at the specified time step $h$,
- `UpdateQbyPV`: add one-step look-ahead value $\sum_{s' \in \mathcal{S}} P(s'|s, a)V_{h+1}(s')$ to $Q_h(s, a)$ for all $(s, a)$ pairs at the specified time step $h$,
- `UpdateV`: take maximum $V_h(s) = \max_{a \in \mathcal{A}} Q_h(s, a)$ for the specified time step $h$,
- `GetQ`: retrieve the values of $Q_h(s, a)$ for all action $a \in \mathcal{A}$ at the specified time step $h$ and state $s$.
- `GetArgMax`: return the indices corresponding to the maximal value in the given list of numbers
- `UpdateQbyBonus`: add exploration bonus to the Q values for all state-action pairs at the specified time step
- `UpdateMDPModel`: update the estimation of the reward and transition function of MDP based on the observed quadruple (old state, action, new state, reward)

The following additional operations are provided to the LLM to emulate MCTS:

- `VisitCheck`: check whether a state has been visited,
- `Rollout`: play the rollout policy until the depth limit $d$ is reached and return the cumulative reward,
- `UCB`: calculate and return the action that maximizes UCB score for the specified state,
- `Generator`: sample the next state and reward based on the transition function,
- `UpdateNandQ`: update the visit count and Q values based on the observed quadruple (old state, action, new state, reward).

Equipped with these operations and the in-context examples showing how to utilize them, STRIDE is capable of computing the optimal policy of MDP with known model by emulating Algorithm 1 and Algorithm 2. Similarly, STRIDE can emulate Algorithm 3 when facing MDP with unknown model, which only needs two additional operations that (i) update the estimation for the unknown reward and transition function, and (ii) update Q values with the exploration bonus, respectively.

### B.2 DYNAMIC PROGRAMMING FOR DYNAMIC MECHANISM DESIGN

For dynamic mechanism design problem, the reference algorithm selected for STRIDE is described in Algorithm 4, which is modified based on the Markov VCG mechanism of Lyu et al. (2022). It is known that the unique solution to equation 3 is the VCG mechanism i.e.,

$$\pi^\star := \arg\max_\pi V^\pi(P, \sum_{i=1}^N \widetilde{R}_i),$$
$$p_i^\star := V^{\pi_{-i}^*}(P, \sum_{j \neq i} \widetilde{R}_j) - V^{\pi^*}(P, \sum_{j \neq i} \widetilde{R}_j), \quad \text{for } i = 1, 2, \ldots, n,$$

where $\pi_{-i}^* := \arg\max_\pi V^\pi(P, \sum_{j \neq i} \widetilde{R}_j)$. Similar to equation 1, equation B.2 can be solved by separately computing policies $\pi^\star$ and $\{\pi_{-i}^*\}_{i=1}^N$ via value iteration, and then evaluating $\pi^\star$ on MDP instances with transition function $P$ and reward function $\sum_{j \neq i} \widetilde{R}_j$ for $i = 1, 2, \ldots, N$.

**Operations.** The following operations are provided to the LLM:

- `UpdateQbyRExcluding`: add immediate rewards, excluding the reward of excluded_agent, to the Q values for all state-action pairs at current time step. If excluded_agent is set to None, all agents' rewards are used.
- `UpdateQbyPVExcluding`: add the one-step look-ahead value, excluding the reward of excluded_agent, to the Q values for all state-action pairs at current time step. If excluded_agent is set to None, all agents' rewards are used.
- `UpdateVExcluding`: update the V values, excluding the reward of excluded_agent, based on the computed Q values for the current time step. If excluded_agent is set to None, all agents's rewards are used.
- `GetQExcluding`: retrieve Q values, that excludes the rewards of excluded_agent, for all actions at the current state and time step. If excluded_agent is set to None, the Q values computed using all agents' rewards will be returned.
- `EvaluatePolicyExcluding`: evaluate the optimal policy on an fictitious MDP that excludes the reward function of excluded_agent.

---

**Algorithm 2** Monte Carlo Tree Search for MDPs.

---

1: **Initialize** $N(s, a) = 0, Q(s, a) = 0, \forall s \in \mathcal{S}, \forall a \in \mathcal{A}$
2: ▷ **Question: Compute the Optimal Policy.**
3: **for** simulation times $n = 1, 2, \cdots, N$ **do**
4:     Observation: Initial state $s_D$
5:     **for** exploration depth $d = D, D - 1, \cdots, 0$ **do**
6:         **if** d = 0 **then**
7:             ▷ Thought: Since the remaining exploration depth is 0. I should update the Nsa values and Q values.
8:         **else**
9:             ▷ Operation: call `VisitCheck` with inputs {state: $s_d$}
10:             **if** $s_d$ has not been visited **then**
11:                 ▷ Thought: We should use rollout policy from now on, until the depth limit $d$ is reached
12:                 ▷ Operation: call `Rollout` with inputs {state: $s_d$, depth: $d$}
13:                 **Break**
14:             **else**
15:                 ▷ Thought: We should search from state $s_d$ by choosing the action that maximizes UCB score.
16:                 ▷ Operation: call `UCB` with inputs {state: $s_d$}
17:                 Observation: Action $a$ maximizes UCB score.
18:                 ▷ Thought: We should simulate playing action $a$ by querying the generator.
19:                 ▷ Operation: call `Generator` with inputs {state: $s_d$, action: $a$}
20:                 Observation: The environment transits to state $s_{d-1}$ and received reward $r_d$
21:     ▷ Thought: We should update the the Nsa values and Q values.
22:     ▷ Operation: call `UpdateNandQ`
23:     Observation: The Nsa values and Q values have been updated.
24: ▷ **Question: which action I should take?**
25: ▷ Thought: I should choose the action that maximizes the computed Q values.
26: ▷ Operation: call `GetQ` with inputs {state: $s_D$}
27: ▷ Operation: call `GetArgMax` with inputs {q_vals: [. . . ]}
28: ▷ Exit: I should choose Action $a$ as it maximizes the Q values. Now exit reasoning.
29: Take action $a = \arg\max_{a \in \mathcal{A}} Q(s_D, a)$

---

- `GetArgMax`: return the indices corresponding to the maximal value in the given list of numbers
- `GetMax`: return the maximal value in the given list of numbers

With these operations, STRIDE is capable of computing the dynamic VCG mechanism by emulating Algorithm 4.

### B.3 BACKWARD INDUCTION FOR BARGAINING IN COMPLETE INFORMATION SETTING

For alternating offer bargaining under complete information, the reference algorithm selected for STRIDE is the backward induction algorithm described in Algorithm 5, which given parameter of the game, including buyer's discount $\delta_b$, seller's discount $\delta_s$, and deadline $T$, can compute the SPE of the game.

**Operations.** The following operations are provided to the LLM:

- `CalcUtil`: calculate buyer or seller's utility using equation 4, with the role of the agent, the specified price and time step as inputs.
- `BackwardOneStep`: compute the SPE price using one step of backward induction reasoning based on the opponent's utility if he/she choose to reject the offer at current time step (see the constrained optimization problem in line 14 and line 17 in Algorithm 5)
- `GetSPEPrice`: retrieve the previously computed SPE price for the specified time step

With these operations, STRIDE is capable of computing the SPE by emulating Algorithm 5. SPE can be used to predict the future offer to be made by the opponent, assuming the opponent is rational

---

**Algorithm 3** Value Iteration Upper Confidence Bound for MDPs with Unknown Model

---

1: **Initialize** $V_{H+1}(s) = 0, \forall s \in \mathcal{S}$
2: **for** episode $t = 1, 2, \dots, T$ **do**
3:     ▷ **Question: Compute the Optimistic Policy for Exploration.**
4:     **for** step $h = H, H-1, \cdots, 1$ **do**
5:         ▷ Thought: Now we can continue to compute the Q-values for the current step $h$.
6:         ▷ Operation: call `UpdateQbyR` with inputs {time_step: h}
7:         ▷ Operation: call `UpdateQbyPV` with inputs {time_step: h}
8:         ▷ Operation: call `UpdateQbyBonus` with inputs {time_step: h}
9:         ▷ Operation: call `UpdateVbyQ` with inputs {time_step: h}
10:         **for** each state $s \in \mathcal{S}$ **do**
11:             **for** each action $a \in \mathcal{A}$ **do**
12:                 ▷ Action: call Python function to calculate Q value for $(s, a)$
13:                 $Q_h(s, a) = \widehat{R}(s, a) + \sum_{s' \in \mathcal{S}} \widehat{P}(s'|s, a) V_{h+1}(s') + b(N(s, a))$
14:         $V_h(s) = \max_{a \in \mathcal{A}} Q_h(s, a)$
15:     ▷ Thought: I have finished value iteration. Now exit reasoning.
16:     **for** step $h = 1, 2, \cdots, H$ **do**
17:         Observe state $s_h$
18:         ▷ **Question: Which action I should take?**
19:         ▷ Thought: I should choose the action that maximizes the computed Q values.
20:         ▷ Operation: call `GetQ` with inputs {time_step: h, cur_state: $s_h$}
21:         ▷ Operation: call `GetArgMax` with inputs {q_vals: [. . .]}
22:         ▷ Exit: I should choose Action $a_h$ as it maximizes the Q values. Now exit reasoning.
23:         Take action $a_h = \arg\max_{a \in \mathcal{A}} Q_h(s_h, a)$
24:         Observe reward $r(s_h, a_h) = R(s_h, a_h) + \epsilon$ and state transits to $s_{h+1}$
25:         ▷ **Question: Update estimations of** $P$ **and** $R$**.**
26:         ▷ Thought: I should update my estimation using the observed $(s_h, a_h, s_{h+1}, r_h)$.
27:         ▷ Operation: call `UpdateMDPModel` with inputs {s: $s_h$, a: $a_h$, s_prime: $s_{h+1}$, r: $r_h$}
28:         ▷ Thought: My estimation is updated. Now exit reasoning.
29:         $N(s_h, a_h) = N(s_h, a_h) + 1$, $N(s_h, a_h, s_{h+1}) = N(s_h, a_h, s_{h+1}) + 1$
30:         $\widehat{P}(s_{h+1}|s_h, a_h) = \frac{N(s_h, a_h, s_{h+1})}{N(s_h, a_h)}$, $\widehat{R}(s, a) = \widehat{R}(s, a) \times \frac{N(s_h, a_h) - 1}{N(s_h, a_h)} + \frac{r(s_h, a_h)}{N(s_h, a_h)}$

---

and that the opponent believes the player to be rational as well. When facing a new offer $p$ made by the opponent at time step $t$, STRIDE will emulate Algorithm 6 to produce a response.

## B.4 BACKWARD INDUCTION FOR BARGAINING IN INCOMPLETE INFORMATION SETTING

Since the seller is uncertain about the value $b$ of the buyer, at each time step $t$ the seller decides the offer price $p_t$ based on his/her belief constructed using observations up to time step $t - 1$, which is denoted as $\mathcal{U}(0, b_{t-1})$, i.e., the true value $b$ is uniformly distributed in $[0, b_{t-1}]$ (with $b_0 = 1$). Therefore, different from SPE considered in complete information setting, SE specifies not only the strategies of the players, but also the belief, which in our case is the sequence $\{b_0, b_1, \dots, b_{T-1}\}$. In classic economics literature (Sobel & Takahashi, 1983; Cramton, 1984), this sequence is obtained by: (i) backward induction from time $T$ to time 1, which results in $b_0$ expressed as a function of $b_{T-1}$; (ii) as the initial belief $b_0 = 1$, we can solve this equation to obtain the value of $b_{T-1}$. This provides an analytical form for $\{b_0, b_1, \dots, b_{T-1}\}$ using the parameters $\delta_b, \delta_s, T$. To make the inner logic more transparent during reasoning, we replace this analytical solution with a bisection search when designing the reference algorithm for STRIDE, with its full description given in Algorithm 7.

We provide the following operations to STRIDE to help it emulate Algorithm 7:

- `CalcUtil`: calculate buyer or seller's utility using equation 5, with the role of the agent, the specified price and time step as inputs.
- `ComputeBt`: compute what seller's belief about buyer's value would be at the current time step, given a guess of seller's belief at time step $T - 1$ (description given in Algorithm 8)
- `SolveLast`: compute seller's expected utility and the corresponding price at the last time step (description given in Algorithm 9)

---

**Algorithm 4** Dynamic VCG Mechanism Design

---

1: **Initialize** $V_{H+1}(s) = 0, V_{H+1,-i}(s) = 0, \forall s \in \mathcal{S}$
2: ▷ **Question: Compute the optimal policy that maximizes all agents' reported rewards.**
3: **for** step $h = H, H-1, \cdots, 1$ **do**
4:     ▷ Thought: Now we can continue to compute the Q-values for the current step $h$.
5:     ▷ Operation: call `UpdateQbyRExcluding` with {time_step: h, excluded_agent:None}
6:     ▷ Operation: call `UpdateQbyPVExcluding` with {time_step: h, excluded_agent:None}
7:     ▷ Operation: call `UpdateVbyQExcluding` with {time_step: h, excluded_agent:None}
8:     **for** each action $a \in \mathcal{A}$ **do**
9:         $Q_h(s,a) = \sum_i^N R_i(s,a) + \sum_{s' \in \mathcal{S}} P(s'|s,a)V_{h+1}(s')$
10:     $V_h(s) = \max_{a \in \mathcal{A}} Q_h(s,a)$
11: ▷ Thought: I have finished value iteration. Now exit reasoning.
12: Denote the optimal policy as $\pi_h^\star(s) := \arg\max_{a \in \mathcal{A}} Q_h(s,a)$ for $h \in [H], s \in \mathcal{S}$
13: **for** step $h = 1, 2, \cdots, H$ **do**
14:     Observe state $s_h$
15:     ▷ **Question: Which action I should take?**
16:     ▷ Thought: I should choose the action that maximizes the computed Q values.
17:     ▷ Operation: call `GetQExcluding` with {time_step: h, cur_state: $s_h$, excluded_agent=None}
18:     ▷ Operation: call `GetArgMax` with {q_vals: [...]}
19:     ▷ Exit: I should choose Action $a_h$ as it maximizes the Q values. Now exit reasoning.
20:     Mechanism designer takes action $a_h = \arg\max_{a \in \mathcal{A}} Q_h(s_h, a)$
21:     Agent $i$ observes reward $r_i(s_h, a_h) = R_i(s_h, a_h) + \epsilon$ for $i \in [N]$ and state transits to $s_{h+1}$
22: **for** agent $i = 1, 2, \cdots, N$ **do**
23:     ▷ **Question: Now compute the VCG price for agent $i$.**
24:     **for** step $h = H, H-1, \cdots, 1$ **do**
25:         ▷ Thought: Now we can continue to compute the Q-values for the current step $h$.
26:         ▷ Operation: call `UpdateQbyRExcluding` with {time_step: h, excluded_agent: $i$}
27:         ▷ Operation: call `UpdateQbyPVExcluding` with {time_step: h, excluded_agent: $i$}
28:         ▷ Operation: call `UpdateVbyQExcluding` with {time_step: h, excluded_agent: $i$}
29:         **for** each state $s \in \mathcal{S}$ **do**
30:             **for** each action $a \in \mathcal{A}$ **do**
31:                 $Q_{h,-i}(s,a) = \sum_{j \neq i} R_j(s,a) + \sum_{s' \in \mathcal{S}} P(s'|s,a)V_{h+1,-i}(s')$
32:             $V_{h,-i}(s) = \max_{a \in \mathcal{A}} Q_{h,-i}(s,a)$
33:     $p_i^\star = V_{1,-i}(s_1) - V^{\pi^\star}(P, \sum_{j \neq i} \widetilde{R}_j)$
34:     ▷ Thought: Now we know the optimal value of this fictitious MDP that ignores agent $i$'s rewards. Next we should evaluate policy $\pi^\star$ on this fictitious MDP.
35:     ▷ Operation: call `EvaluatePolicyExcluding` with {excluded_agent: $i$}
36:     ▷ Thought: Then the VCG price for agent $i$ is simply their difference ... Now exit reasoning.

---

- `Solve`: compute the expected utility and the corresponding price at the current time step, based on the results computed for the next time step (description given in Algorithm 10)
- `GetSEPrice`: retrieve the previously computed SE price for the specified time step

Then similar to the complete information setting, when deciding whether to accept an offer from the seller, the buyer can compare the utility he/she can get by accepting the current offer, and the utility he/she can get by waiting for seller's offer in the next time step. For the latter, as the buyer assumes the seller is rational, the next offer from seller is predicted using the SE price from Algorithm 7.

## C    PROMPTS OF THE STRIDE FRAMEWORK AND BASELINES

The prompts used for the LLM agents in Section 4 consist of three parts, which we mark using different colors in this section: a system prompt setting the role of the agent (gray), followed by a formal description of the decision-making problem to be solved (light blue), and then parameters of the problem instance (light green). The system prompt is problem-agnostic, which is given below.

---

**Algorithm 5** Backward Induction to Compute SPE of Bargaining under Complete Information

1: ▷ **Question: Compute the SPE Prices via Backward Induction.**
2: **for** time step $t = T, T-1, \cdots, 1$ **do**
3:     ▷ Thought: Compute the SPE price for time $t$, based on the results computed for time $t+1$
4:     **if** $t = T$ **then**
5:         **if** current_player = Buyer **then**
6:             ▷ Operation: call BackwardOneStep with {agent: buyer, op_u: 0.0, t: $T$}
7:             The SPE price $p_T := 0.0$
8:         **else**
9:             ▷ Operation: call BackwardOneStep with {agent: seller, op_u: 0.0, t: $T$}
10:            The SPE price $p_T := 1.0$
11:    **else**
12:        **if** current_player = Buyer **then**
13:            ▷ Operation: call BackwardOneStep with {agent: buyer, op_u: $u_s(p_{t+1}, t+1)$, t: $t$}
14:            The SPE price $p_t := \arg\max_p u_b(p, t)$, s.t. $u_s(p, t) \geq u_s(p_{t+1}, t+1)$
15:        **else**
16:            ▷ Operation: call BackwardOneStep with {agent: seller, op_u: $u_b(p_{t+1}, t+1)$, t: $t$}
17:            The SPE price $p_t := \arg\max_p u_s(p, t)$, s.t. $u_b(p, t) \geq u_b(p_{t+1}, t+1)$.
18:    ▷ Operation: call CalcUtil with {agent: seller, price: $p_t$, t: $t$}
19:    ▷ Operation: call CalcUtil with {agent: buyer, price: $p_t$, t: $t$}
20:    Buyer utility $u_b(p_t, t)$, Seller utility $u_s(p_t, t)$
21: ▷ Thought: SPE prices for all time steps are calculated. Now exit reasoning.

---

**Algorithm 6** Response to Offer in Bargaining with Complete Information

1: **Inputs:** current_player, price $p$, time $t$, SPE prices $\{p_t\}_{t=1}^T$
2: ▷ **Question: Should I accept or reject opponent's offer?**
3: ▷ Thought: I should first compute the utility I get by accepting the offer, and then the utility I
    get by rejecting the offer and making a counter offer using the SPE price in the next time step.
4: ▷ Operation: call CalcUtil with inputs {agent: current_player, price: $p$, t: $t$}
5: ▷ Operation: call GetSPEPrice with inputs {t: $t+1$}
6: ▷ Operation: call CalcUtil with inputs {agent: current_player, price: $p_{t+1}$, t: $t+1$}
7: **if** current_player = buyer **then**
8:     $u_a = u_b(p, t)$, $u_r = u_b(p_{t+1}, t+1)$
9: **else**
10:    $u_a = u_s(p, t)$, $u_r = u_s(p_{t+1}, t+1)$
11: **if** $u_a \geq u_r$ **then**
12:    ▷ Thought: I should accept the offer. Now exit reasoning.
13:    **return** Accept
14: **else**
15:    ▷ Thought: I should reject the offer. Now exit reasoning.
16:    **return** Reject

---

**System prompt for zero-shot CoT**

You are a world class intelligent agent capable of solving various classes of decision making problems. For each decision making problem you encounter next, you will be given the description of the problem setup and your objective. You need to carefully reason about the problem step-by-step, and make optimal decisions for the encountered problem instance.

---

**System prompt for zero-shot CoT w/ code interpreter**

---

**Algorithm 7** Backward Induction to Compute SE of Bargaining under Incomplete Information

---

1: ▷ **Question: Compute the SE Prices via Bisection Search and Backward Induction.**

2: ▷ Thought: I need to first compute my belief about buyer's value at time step T-1 under sequential equilibrium, denoted $b_{T-1}$, which can be done via bisection search. I should terminate when the value $b'_0$ computed based on $b'_{T-1}$ gets close enough to my actual initial belief $b_0 = 1.0$.

3: $l = 0, h = 1, B'_{T-1} = (l + h)/2$

4: ▷ Operation: Call $\texttt{ComputeBt}$ with inputs {time_step: 1, b_last: $B'_{T-1}$}

5: $b'_0 = \texttt{ComputeBt}(1, b'_{T-1})$

6: **while** $|b'_0 - 1.0| \geq 10^{-3}$ **do**

7:     **if** $b'_0 \leq 1.0$ **then**

8:         ▷ Thought: Since $b'_0$ is smaller than $b_0$, I should focus on the region $[b'_{T-1}, h]$ next time.

9:         $l = b'_{T-1}$

10:     **else**

11:         ▷ Thought: Since $b'_0$ is larger than $b_0$, I should focus on the region $[l, b'_{T-1}]$ next time.

12:         $h = b'_{T-1}$

13:     $b'_{T-1} = (l + h)/2$

14:     ▷ Operation: Call $\texttt{ComputeBt}$ with inputs {time_step: 1, b_last: $B'_{T-1}$}

15:     $b'_0 = \texttt{ComputeBt}(1, b'_{T-1})$

16: ▷ Thought: Since $|b'_0 - 1.0| < 10^{-3}$, the value of my initial belief computed based on $B'_{T-1}$ is close enough to the actual value $b_0 = 1$. Therefore, $B'_{T-1}$ is an accurate approximation of $B_{T-1}$ in SE. Now I can start backward induction to compute the SE prices from time $T$ to 1.

17: **for** $t = T, T - 1, \ldots, 1$ **do**

18:     **if** $t = T$ **then**

19:         ▷ Operation: Call function SOLVELAST with inputs {b_last: $B'_{T-1}$}.

20:         $u_t, p_t = \texttt{SolveLast}(B'_{T-1})$ # seller's expected utility and price under SE

21:     **else**

22:         ▷ Operation: Call function SOLVE with inputs {u: $u_{t+1}$, p: $p_{t+1}$, t: $t$}.

23:         $u_t, p_t = \texttt{Solve}(u_{t+1}, p_{t+1}, t)$ # seller's expected utility and price under SE

24:     ▷ Thought: Now I need to continue to time step $t - 1$.

25: ▷ Thought: I have reached $t = 1$. Exit reasoning now.

---

**Algorithm 8** $\texttt{ComputeBt}$

---

1: **Inputs** time_step, the time index of current belief, and b_last, the belief at time step $T$.

2: **Initialize** constants $\{c_\tau\}_{\tau=2}^T$ with $c_T = 0.5$ and $c_\tau = \frac{(1-\delta_b+\delta_b c_{\tau+1})^2}{2(1-\delta_b+\delta_b c_{\tau+1})-\delta_s c_{\tau+1}}$ for $\tau \geq 2$.

3: Set $t = $ time_step, $b_{T-1} = $ b_last

4: **for** $\tau = T - 1, T - 2, \ldots, t$ **do**

5:     $b_{\tau-1} = \frac{2(1-\delta_b+\delta_b c_{\tau+1})-\delta_s c_{\tau+1}}{1-\delta_b+\delta_b c_{\tau+1}} b_\tau$

6: **return** $b_{t-1}$

---

You are a world class intelligent agent capable of solving various classes of decision making problems. For each decision making problem you encounter next, you will be given the description of the problem setup and your objective. You need to carefully reason about the problem step-by-step, and make optimal decisions for the encountered problem instance. You are provided with a code interpreter. You should write and run code to answer the questions.

**System prompt for few-shot CoT w/ code interpreter**

You are a world class intelligent agent capable of solving various classes of decision making problems. For each decision making problem you encounter next, you will be given the de-

---

**Algorithm 9** `SolveLast`

---

1: **Inputs** b_last, the belief at time step $T$.
2: Set $b_{T-1} = $ b_last
3: Compute SPE price $p_T := \arg\max_p p \cdot \frac{b_{T-1}-p}{b_{T-1}} = \frac{1}{2} b_{T-1}$
4: Compute expected utility $u_T := p_T \cdot \frac{b_{T-1}-p_T}{b_{T-1}} = \frac{1}{4} b_{T-1}$
5: **return** $u_T, p_T$

---

**Algorithm 10** `Solve`

---

1: **Inputs** u, seller's expected utility at t+1, p, the associated price, and t, the current time step.
2: Set $u_{t+1} = $ u, $p_{t+1} = $ p, and $t = $ t
3: Compute SPE price

$$p_t := \arg\max_p \frac{b_{t-1}-b_t}{b_{t-1}} p + \frac{b_t}{b_{t-1}} u_{t+1}, \text{ s.t. } b_t = \delta_b(b_t - p_{t+1})$$

$$= (1 - \delta_b)b_t + \delta_b p_{t+1}$$

4: Compute expected utility $u_t = \frac{b_{t-1}-b_t}{b_{t-1}} p_t + \frac{b_t}{b_{t-1}} u_{t+1}$
5: **return** $u_t, p_t$

---

scription of the problem setup and your objective. Your need to carefully reason about the problem, and make optimal decisions for the encountered problem instance. You are provided with a code interpreter and an example implementation. You should write and run code to answer the questions following the example.

---

**System prompt for STRIDE**

You are a world class intelligent agent capable of solving various classes of decision making problems. For each decision making problem you encounter next, you will be given the description of the problem setup and your objective. Your need to carefully reason about the problem step-by-step, and make optimal decisions for the encountered problem instance. You are provided with a set of tools that handle low-level calculations and examples showing you how to use these tools to solve this problem.

---

In the remainder of this section, we will provide the prompts describing the decision making problems and the problem parameters to the agents.

C.1 MDP WITH KNOWN MODEL

The following are the prompts we provide to all agents to describe the formulation and the agent's objective in MDP when the model, i.e., the transition function and reward function, is known.

---

**Description of MDP with known model**

A finite horizon tabular Markov Decision Process (MDP) is a model for decision-making in scenarios where outcomes are influenced by both randomness and controlled decisions, with decisions being made over a finite number of time steps.

Components:

State Space $S$: $s_0, s_1, \ldots, s_{|S|-1}$, where $|S|$ is the total number of states.

Action Space $A$: $a_0, a_1, \ldots, a_{|A|-1}$, where $|A|$ is the total number of actions.

---

Transition probability matrix $P$: a three-dimensional tensor with shape $|S| \times |A| \times |A|$, where each entry represents the probability of transitioning from one state after taking a specific action to another state.

Reward matrix $R$: a matrix with shape $|S| \times |A|$, where each entry gives the mean of the immediate reward received after taking an action in a state.

Horizon length $H$: The total number of time steps the decision process is constrained to.

Interaction protocol:

For time step $h = 1, 2, \ldots, H$

Agent takes an action $a_h \in A$ based on the current state $s_h$

Agent receives reward $r_h := R[s_h, a_h] + \eta_h$, where $\eta_h \sim \mathcal{N}(0, 1)$

The environment transits to the next state $s_{h+1}$ with probability $P[s_h, a_h, s_{h+1}]$ Goal of the agent:

Maximize expected cumulative rewards $\mathbb{E}\left[\sum_{h=1}^{H} R[s_h, a_h]\right]$, where the expectation is w.r.t. randomness of agent's policy and state transition.

For zero-shot CoT, which can only read the parameters from context, we print the complete transition matrix $P$ and reward matrix $R$ as shown below, where the empty curly brackets {} are substituted with actual values of the problem instance.

| **Description of problem instance** |
| --- |
| Now you are going to play in a finite-horizon tabular Markov decision process, with length of horizon {} (with time indices starting from h=0 to {}), number of states —S—={}, number of actions —A—={}. The transition matrix P is: {} and reward matrix R is {}. |

For zero-shot CoT w/ code, few-shot CoT w/ code and STRIDE, which can read the parameters from their working memory or an external file, instead of directly printing the transition and reward matrices in context, we state in the prompt where these values can be accessed.

| **Description of problem instance** |
| --- |
| Now you are going to play in a finite-horizon tabular Markov decision process, with length of horizon {} (with time indices starting from h=0 to {}), number of states —S—={}, number of actions —A—={}. The transition matrix P and reward matrix R are stored in working memory. |

## C.2 MDP WITH UNKNOWN MODEL

The following are the prompts we provide to all agents to describe the formulation and the agent's objective in MDP when the model, i.e., the transition function and reward function, is unknown.

| **Description of MDP with unknown model** |
| --- |
| A finite horizon tabular Markov Decision Process (MDP) is a model for decision-making in scenarios where outcomes are influenced by both randomness and controlled decisions, with decisions being made over a finite number of time steps.

Components: |

State Space $S$: $s_0, s_1, \ldots, s_{|S|-1}$, where $|S|$ is the total number of states.

Action Space $A$: $a_0, a_1, \ldots, a_{|A|-1}$, where $|A|$ is the total number of actions.

Transition probability matrix $P$: a three-dimensional tensor with shape $|S| \times |A| \times |A|$, where each entry represents the probability of transitioning from one state after taking a specific action to another state.

Reward matrix $R$: a matrix with shape $|S| \times |A|$, where each entry gives the mean of the immediate reward received after taking an action in a state.

Horizon length $H$: The total number of time steps the decision process is constrained to.

Number of episodes $K$: The total number episodes the MDP is repeatedly played by the agent, where in each episode, the agent starts fresh, makes a series of $H$ decisions and then the episode ends. Note that learning achieved in earlier episodes influences the behavior in later episodes. Unknown model of the environment: The transition probability matrix $P$ and reward matrix $R$ are unknown to the agent, and the agent needs to estimate them based on the collected observations and improve its policy after each episode.

Interaction protocol:

For episode $k = 0, 1, 2, \ldots, K - 1$:

For time step $h = 0, 1, 2, \ldots, H - 1$:

Agent takes an action $a_{k,h} \in A$ based on the current state $s_{k,h}$

Agent receives reward $r_{k,h} := R[s_{k,h}, a_{k,h}] + \eta_{k,h}$, where $\eta_{k,h} \sim \mathcal{N}(0, 1)$

The environment transits to the next state $s_{k,h+1}$ with probability $P[s_{k,h}, a_{k,h}, s_{k,h+1}]$

Agent can update its estimation of matrix $P$ and $R$ based on the newly observed quadruples $(s_{k,h}, a_{k,h}, s_{k,h+1}, r_{k,h+1})$ for $h = 0, 1, 2, \ldots, H - 1$

Goal of the agent:

Maximize expected cumulative rewards $E\left[\sum_{k=0}^{K-1} \sum_{h=0}^{H-1} R[s_h, a_h]\right]$, where the expectation is w.r.t. randomness of agent's policy and state transition.

For STRIDE, since it can automatically update, store, and read the estimated transition and reward matrices in working memory, we simply use the following description about the problem instance for all episodes.

### Description of problem instance

Now you are going to play in a finite-horizon tabular Markov decision process, with length of horizon {} (with time indices starting from h=0 to {}), number of states —S—={}, number of actions —A—={}. The transition matrix P and reward matrix R are unknown to you, so you need to estimate them based on interaction history.

For all the baselines, since they cannot reliably summarize the interaction history and construct the estimation of $P$ and $R$, we explicitly provide the estimation of $P$ and $R$ and the count of visitation of state-action pairs as shown below. This is similar to the "externally summarized interaction history" in the prompt for multi-armed bandit problems used by Krishnamurthy et al. (2024).

### Description of problem instance

Now you are going to play in a finite-horizon tabular Markov decision process, with length of horizon {} (with time indices starting from h=0 to {}), number of states —S—={}, number

of actions —A—={}. The transition matrix P and reward matrix R are unknown to you. Your current estimation of transition matrix P is {}, your current estimation of reward matrix R is {}, and your count of visitation of all the state-action pairs is {}.

### C.3 DYNAMIC MECHANISM DESIGN PROBLEM

The following are the prompts we provide to all agents to describe the formulation and the agent's objective in Dynamic Mechanism Design problem, when the model, i.e., the transition function and reward function, is known.

---

**Description of dynamic mechanism design problem**

The dynamic mechanism design problem involves creating allocation and pricing rules for decision-making, where the value of resource to the agents changes over time as the state of the environment changes.

Components:

Players: a mechanism designer and a set of $N$ agents State Space $S$: $s_0, s_1, \ldots, s_{|S|-1}$, where $|S|$ is the total number of states.

Action Space $A$: $a_0, a_1, \ldots, a_{|A|-1}$, where $|A|$ is the total number of actions. Each action represents the mechanism designer's allocation of some scarce resource among $N$ agents.

Transition probability matrix $P$: a three-dimensional tensor with shape $|S| \times |A| \times |A|$, where each entry represents the probability of transitioning from one state after taking a specific action to another state.

Reward matrix $R$: a three-dimensional tensor with shape $N \times |S| \times |A|$, where each matrix $R[i,:,:]$ represents the reward matrix of an agent $i$ for $i = 1, 2, \ldots, N$, and each of its entry gives the mean of the immediate reward received by agent $i$ after the mechanism designer takes an action in a state.

Horizon length $H$: The total number of time steps the decision process is constrained to.

Interaction protocol:

Before the interaction starts, each agent $i$ reports a reward matrix (can be different from its true reward matrix $R[i,:,:]$), denoted as $\widetilde{R}[i,:,:]$, to the designer. Based on agents' reported reward matrix, the designer chooses a policy $\pi : S \to \Delta(A)$ and prices $\{p_i\}_{i=1}^N$ to be charged to each agent.

For time step $h = 1, 2, \ldots, H$:

Mechanism designer takes an action $a_h \sim \pi(s_h)$ based on the policy $\pi$ and the current state $s_h$

Each agent $i$ receives reward $R[i, s_h, a_h]$ for $i = 1, 2 \ldots, N$ The environment transits to the next state $s_{h+1}$ with probability $P[s_h, a_h, s_{h+1}]$

After the interaction, the mechanism designer charges each agent $i$ with some price $p_i$

Goal of the agents:

Each agent wants to maximize its utility $u_i = \mathbb{E}\left[\sum_{h=1}^H R[i, s_h, a_h]\right] - p_i$, that is, the difference

between the expected cumulative rewards, where the expectation is w.r.t. randomness of designer's policy and state transition, and the price charged by the mechanism designer. As the agents cannot directly take actions, their only leverage is to decide whether to truthfully report their reward matrix to the designer.

Goal of the mechanism designer:

---

Maximize the expected cumulative rewards of all agents $E\left[\sum_{i=1}^{N}\sum_{h=1}^{H}R[i,s_h,a_h]\right]$, where the expectation is w.r.t. randomness of designer's policy and state transition. As the designer only observes agents' reported reward matrix $\widetilde{R}$, to fulfil its objective, the designer needs to guarantee, with its policy and pricing strategy, no agent $i$ has incentive to report $\widetilde{R}[i,:,:]$ that is different from the true reward matrix $R[i,:,:]$ unilaterally.

It is known that VCG mechanism guarantees truthfulnes of the agents, and uniquely maximizes the objective. It is defined as follows:

$$\pi^\star = \arg\max_{\pi}\mathbb{E}_{\pi,P}\left[\sum_{i=1}^{N}\sum_{h=1}^{H}\widetilde{R}[i,s_h,a_h]\right]$$

$$p_i^\star = \mathbb{E}_{\pi_{-i}^\star,P}\left[\sum_{j\neq i}\sum_{h=1}^{H}\widetilde{R}[j,s_h,a_h]\right] - \mathbb{E}_{\pi^\star,P}\left[\sum_{j\neq i}\sum_{h=1}^{H}\widetilde{R}[j,s_h,a_h]\right]$$

for $i = 1, 2, \ldots, N$, where $\pi_{-i}^\star = \arg\max_{\pi}\mathbb{E}_{\pi,P}\left[\sum_{j\neq i}\sum_{h=1}^{H}\widetilde{R}[j,s_h,a_h]\right]$ is the optimal policy for a MDP with transition probability matrix P and reward matrix $\sum_{j\neq i}\widetilde{R}[j,:,:]$, that is, excluding the reward matrix of agent $i$ itself.

Now as a strategic decision maker, your job is to compute the VCG mechanism based on the given transition probability matrix $P$ and the reward matrix $R$ reported by the agents. Then you should take an action at each time step and charges prices to each agent at the end, according to your computed VCG mechanism.

---

**Description of problem instance**

Now you are going to play in a finite-horizon dynamic mechanism design problem, with number of agents N={}, length of horizon {} (with time indices starting from h=0 to {}), number of states —S—={}, number of actions —A—={}. The transition matrix P is:{} and reward matrix R reported by the agents is {}.

---

C.4   SINGLE-ISSUE BARGAINING UNDER COMPLETE INFORMATION

The following are the prompts we provide to all agents to describe the formulation and the agent's objective in single-issue bargaining under complete information.

---

**Description of single-issue bargaining under complete information**

The alternating offer bargaining game is a negotiation framework between two players, a buyer and a seller, aimed at determining the price of an item. This strategic game plays out over several rounds with a finite deadline, emphasizing the tactics of bargaining under time constraints.

Components:

Players: Two (Buyer and Seller).

Buyer's Value: 1 (the maximum price the buyer is willing to pay). Seller's Value: 0 (the minimum price the seller is willing to accept).

Discount Factors ($\delta_b$ and $\delta_s$): Represents how much each player values immediate transactions over future possibilities, where $\delta_b, \delta_s \in (0,1)$. Utility associated with future offers are discounted by $\delta_b^{t-1}$ and $\delta_s^{t-1}$ for the buyer and the seller, respectively, where t indicates the current round.

Buyer's Utility: If a price $p$ is agreed upon at time step $t <= T$, then buyer's utility is $u_b = (1 - p) * \delta_b^{t-1}$.

Seller's Utility: If a price $p$ is agreed upon at time step $t <= T$, then seller's utility is $u_b = (p - 0) * \delta_s^{t-1}$.

Deadline: If no sale is agreed upon by the end of time T, the negotiation fails, and no transaction occurs, in which case, both agents get 0 utility.

Complete Information: All details about the item's value range, the structure of the rounds, and the potential outcomes are common knowledge.

Interaction Protocol:

Decision Turns: Starting with the buyer, players alternate in making price offers. The player making an offer proposes a price within the range from the seller's value to the buyer's value.

Responses: The opponent can either accept the proposed price, resulting in a sale and the game ending, or reject the offer, in which case the negotiation advances to the next round.

Goal of the agents:

The seller aims to maximize the sale price while the buyer seeks to minimize it. Each agent's goal is to negotiate a price as close as possible to their value (1 for the seller, 0 for the buyer) while considering the risk of no agreement by the deadline.

---

**Description of problem instance**

# For buyer

This is the beginning of a new game instance, where you will play as the buyer. Your discount factor $\delta_b$={}, seller's discount factor $\delta_s$={}, and the deadline T={}. In the following, you should make your decision by assuming your opponent is rational as well.

# For seller

This is the beginning of a new game instance, where you will play as the seller. Your discount factor $\delta_s$={}, buyer's discount factor $\delta_b$={}, and the deadline T={}. In the following, you should make your decision by assuming your opponent is rational as well.

---

C.5 SINGLE-ISSUE BARGAINING UNDER INCOMPLETE INFORMATION

The following are the prompts we provide to all agents to describe the formulation and the agent's objective in single-issue bargaining under incomplete information.

---

**Description of single-issue bargaining under incomplete information**

This is a finite horizon bargaining game with one-sided uncertainty, in which the uninformed bargainer, the seller, makes all the offers and the informed bargainer, the buyer, can only decides to accept or reject the offer.

Components:

Players: Buyer (informed) and Seller (uninformed).

Buyer's Value: b (the maximum price the buyer is willing to pay).

Seller's Value: 0 (the minimum price the seller is willing to accept).

Discount Factors ($\delta_b$ and $\delta_s$): Represents how much each player values immediate transactions over future possibilities, where $\delta_b, \delta_s \in (0, 1)$. Utility associated with future offers are

discounted by $\delta_b^{t-1}$ and $\delta_s^{t-1}$ for the buyer and the seller, respectively, where t indicates the current time step.

Buyer's Utility: If a price $p$ is agreed upon at time step $t <= T$, then buyer's utility is $u_b = (b - p) * \delta_b^{t-1}$.

Seller's Utility: If a price $p$ is agreed upon at time step $t <= T$, then seller's utility is $u_b = (p - 0) * \delta_s^{t-1}$.

Deadline: If no sale is agreed upon by the end of time T, the negotiation fails, and no transaction occurs, in which case, both agents get 0 utility.

Information Asymmetry: Buyer himself knows the true value of b, which is drawn from a known distribution $F(v)$ supported on $[0, 1]$. We assume $F(v) = v$, i.e., Buyer's value $b$ is sampled from a uniform distribution. The seller does not know $b$ but knows the distribution $F(v)$.

Interaction Protocol:

Decision Turns: In each time step $t = 1, 2, \ldots, T$, it is always Seller who makes an offer $p_t$ within the range of [0,1].

Responses: Buyer can either accept the proposed price, resulting in a sale and the game ending, or reject the offer, in which case the negotiation advances to the next time step.

Goal of the agents:

Seller's Objective: Maximize their expected payoff over the horizon of the game without knowing the true value of $b$. The seller must strategically decide on the prices $p_t$ to offer in each time step, considering the declining number of opportunities to make a sale and the distribution of $b$ inferred from the buyer's responses.

Buyer's Objective: Maximize their surplus, which is the difference between the true value $b$ and the price paid $p$, if a transaction occurs. The buyer needs to decide whether to accept or reject the seller's offers based on the value $b$ and the likelihood of a more favorable price in subsequent time steps, considering the finite number of time steps.

---

**Description of problem instance**

# For buyer

This is the beginning of a new game instance, where you will play as the buyer. Your discount factor $\delta_b$={}, seller's discount factor $\delta_s$={}, and the deadline T={}. Your value $b = \{\}$, which is uniformly sampled from $[0, 1]$. In the following, you should make your decision by assuming your opponent is rational as well.

# For seller

This is the beginning of a new game instance, where you will play as the seller. Your discount factor $\delta_s$={}, buyer's discount factor $\delta_b$={}, and the deadline T={}. The buyer's value $b$ is unknown to you, but you know it is uniformly sampled from $[0, 1]$. In the following, you should make your decision by assuming your opponent is rational as well.

---

## C.6 TIC-TAC-TOE

The following are the prompts we provide to all agents to describe the formulation and the agent's objective for the Tic-Tac-Toe game. The prompts also detail the agents' goals and initial game setup.

---

**Description of Tic-Tac-Toe Game**

Tic-Tac-Toe is a classic two-player game where players take turns marking spaces in a 3x3 grid, aiming to place three of their marks in a horizontal, vertical, or diagonal row to win.
Components:
- Players: Two players, usually denoted as Player X and Player O.
- Board: A 3x3 grid where each cell can be empty, marked with an X, or marked with an O.
- Marks: Each player has a unique mark (X or O) that they place on the board.

Interaction Protocol:
- Players take turns starting with Player X.
- On each turn, a player marks an empty cell on the grid with their mark (X or O).
- The game continues until a player has three of their marks in a horizontal, vertical, or diagonal row, or all cells are filled resulting in a draw.

Rules:
1. Players alternate turns, with Player X always going first.
2. A player can only mark an empty cell.
3. The game ends when one player achieves a row of three marks horizontally, vertically, or diagonally, or when all cells are filled with no winner (a draw).

Goals of the Players:
- Player X: Maximize the chances of placing three X's in a row before Player O does.
- Player O: Maximize the chances of placing three O's in a row before Player X does.

Winning Conditions:
- A player wins if they place three of their marks in a horizontal, vertical, or diagonal row.
- If all cells are filled without any player achieving three marks in a row, the game results in a draw.

Game Setup:
1. The game begins with an empty 3x3 grid.
2. Players decide who will be Player X and who will be Player O.
3. Player X makes the first move.

Objective:
Each player aims to either achieve a row of three of their marks or to block the opponent from doing so. Strategic planning and anticipation of the opponent's moves are crucial to winning the game.

---

**Description of problem instance**

Now you are going to play a game of Tic-Tac-Toe. The current state of the board is {}. It is player {}'s turn. Your objective is to place three of your marks in a horizontal, vertical, or diagonal row to win while preventing your opponent from doing the same.

---

## C.7 CONNECT-N

The following are the prompts we provide to all agents to describe the formulation and the agent's objective for the Connect-N game. The prompts also detail the agents' goals and initial game setup.

---

**Description of Connect-N**

Connect-N is a generalized version of Connect-4, where two players alternate turns dropping colored discs into a vertically suspended grid. The objective is to form a horizontal, vertical, or diagonal line of $N$ discs. The game introduces a gravity effect where discs drop to the lowest available position within a column, adding a unique strategic dimension to the gameplay.

Components:

- Players: Two players, typically referred to as Player X and Player O, who use different colored discs.
- Board: A grid with configurable dimensions, larger than the typical 3×3 Tic-Tac-Toe board.
- Discs: Each player has an ample supply of discs in their respective colors.

Interaction Protocol:

- Players take turns, starting with Player X.
- On each turn, a player chooses a column to drop a disc into. The disc falls, affected by gravity, to the lowest available position within the column.
- The game continues until a player forms a line of N discs in a row (horizontally, vertically, or diagonally) or the board is completely filled, resulting in a draw.

Rules:

1. Players must alternate turns, with Player X always going first.
2. A player can only choose a column that has available space.
3. The game ends when one player forms a line of N discs or when all columns are filled without any player achieving this, which results in a draw.

Goals of the Players:

- Player X: Strategize to connect N of their discs in a row vertically, horizontally, or diagonally before Player O.
- Player O: Similarly, aim to connect N of their discs in a row while blocking Player X's attempts.

Winning Conditions:

- A player wins by aligning N of their discs in a row in any direction.
- The game results in a draw if the entire board is filled without either player achieving N in a row.

Game Setup:

1. The game starts with an empty board of the chosen dimensions.
2. Players decide who will play first (Player X) and choose their disc colors.
3. Player X makes the first move by dropping a disc into one of the columns.

Objective:

Each player aims to strategically drop their discs to form a line of $N$ while preventing their opponent from doing the same. Anticipating the opponent's moves and effectively using the gravity-affected game-play are critical to securing a victory.

---

**Description of problem instance**

---

Now, you are going to play a game of Connect-N, where two players alternate turns dropping discs into a vertically suspended grid. The objective is to form a line of $N$ discs in a row, either horizontally, vertically, or diagonally. The current state of the board is {}, the current player is Player {}, the number of discs required to win is {}. Your objective is to strategically drop your discs to form a line of {} discs while preventing your opponent from doing the same.

---

Description of Problem Instance Current board: {self.board}, Player: {self.player}, Available moves: {self.get_available_moves()}

---

## D   ADDITIONAL EXPERIMENTS

In this section, we further evaluate STRIDE on a set of additional experiments.

---

**Algorithm 11** BFS Minimax with Alpha-Beta Pruning

---

1: **function** BFSALPHABETA($root$, $\alpha$, $\beta$)
2:     $queue \leftarrow$ new Queue()
3:     $parentMap \leftarrow$ new Dictionary()                 $\triangleright$ To store parent-child relationships
4:     $queue$.enqueue($\{root, \alpha, \beta\}$)
5:     $scores \leftarrow$ new Dictionary()                       $\triangleright$ To store scores temporarily
6:     **while** $queue$ is not empty **do**
7:         $\{node, current\_alpha, current\_beta\} \leftarrow queue$.dequeue()
8:         **if** $node$ is a terminal state **then**
9:             $scores[node] \leftarrow U(node)$                   $\triangleright$ Utility of terminal state
10:        **else**
11:            $value \leftarrow -\infty$ if $node$.isMaximizingPlayer() else $\infty$
12:            **for all** $child \in$ Children($node$) **do**
13:                $queue$.enqueue($\{child, current\_alpha, current\_beta\}$)
14:                $parentMap[child] \leftarrow node$
15:         **if** node in $parentMap$ **then**
16:            $parent \leftarrow parentMap[node]$
17:            $eval \leftarrow scores[node]$
18:            **if** $parent$.isMaximizingPlayer() **then**
19:                $scores[parent] \leftarrow \max(scores[parent], eval)$
20:                $current\_alpha \leftarrow \max(current\_alpha, scores[parent])$
21:            **else**
22:                $scores[parent] \leftarrow \min(scores[parent], eval)$
23:                $current\_beta \leftarrow \min(current\_beta, scores[parent])$
24:         **if** $current\_beta \leq current\_alpha$ **then**
25:            **break**                               $\triangleright$ Pruning
26:     **return** $scores[root]$

---

## D.1   TIC-TAC-TOE AND CONNECT-N

Here we evaluate STRIDE and the baselines (`GPT-3.5-Turbo-0125` with the temperature set to 0) on Tic-Tac-Toe and Connect-N Games. For these two games, we provide STRIDE with tools and demonstration that make it emulate Minimax algorithm as shown in Algorithm 11.

**Agent's Objective in Tic-Tac-Toe.** The primary objective for each agent is to win the Tic-Tac-Toe game by placing three markers in the same row, column, or diagonal before the opponent. If a win is not feasible, the secondary objective is to aim for a tie, preventing the opponent from winning. Each agent strives to select the optimal action based on the game's current state. If both players play optimally, the game results in a tie.

**Experiment Setup and Results.** In addition to the baselines mentioned in Section 4, here we also include *RAFA with Monte Carlo Tree Search* (MCTS) (Liu et al., 2023) and *RAFA with Minimax*. For *CoT w/ code*, the LLM has been instructed to implement Minimax algorithm to play the game, and for the *RAFA* agents, the search breadth, denoted $B$, is set to 4. In addition to the original *RAFA MCTS* implementation[1], we implemented *RAFA with Minimax* as an extra baseline. We adopt the memory structure from their original implementation to store optimal actions and use similar prompts and interactions with the LLM to expand the game tree and assess game states. Additionally, for *RAFA with Minimax*, we set the search depth, denoted $U$, to the maximum value 9.

In our experiments, STRIDE is equipped with operational tools to emulate a Breadth-First version of Minimax algorithm with alpha-beta pruning (see Algorithm 11). Starting from depth 0 and progressing to the maximum depth — determined by the total number of empty cells on the board — the algorithm evaluates potential outcomes at each node: $+1$ for a win, $-1$ for a loss, and 0 for a tie or non-terminal states. Utilizing backward induction, the algorithm recursively refines and updates these scores, ensuring that the decision path optimizes the expected outcome at each node from the current player's perspective. These scores are stored in STRIDE's working memory. When STRIDE

---

[1]https://github.com/agentification/RAFA_code

agent starts playing the game, it retrieves the scores for each possible action, and then selects the action with maximal or minimal score depending on the role of the player. We repeat the experiments on a fixed set of parameters for 10 runs, with the initial player being 'X' and an empty board to start the game. The results are presented in Table 7.

Table 7: Model performances in Tic-Tac-Toe (10 runs).

| Outcome | RAFA w/ Minimax | RAFA w/ MCTS | zero-shot CoT | zero-shot CoT w/ code | STRIDE |
|---|---|---|---|---|---|
| X Wins (%) | 50 | 60 | 70 | 80 | 20 |
| Tie (%) | 30 | 20 | 0 | 20 | **80** |
| O Wins (%) | 20 | 20 | 30 | 0 | 0 |

**STRIDE Vs. Baseline Models**  We also conducted experiments that pit STRIDE against baseline models in Tic-Tac-Toe, including *zero-shot CoT*, *zero-shot CoT w/ code*, and *RAFA w/ MCTS*. We instructed *zero-shot CoT w/ code* to implement the Minimax algorithm, and for *RAFA w/ MCTS*, we set $B = 4$ and $U = 4$. The experiments were conducted over 10 runs, with STRIDE playing as player 'X' and the baseline models as player 'O'. The outcomes are summarized in Table 8.

Table 8: STRIDE against Baseline Models in Tic-Tac-Toe (10 runs)

| Matchup | STRIDE Wins (%) | Tie (%) | Opponent Wins (%) |
|---|---|---|---|
| STRIDE vs zero-shot CoT | **90** | 10 | 0 |
| STRIDE vs zero-shot CoT w/ code | **80** | 20 | 0 |
| STRIDE vs RAFA w/ MCTS | **50** | 50 | 0 |

**Agent's Objective in Connect-N.** In Connect-N, available moves can be made in the lowest empty space of each column.The agent aims to drop its discs to form a line of $N$ while preventing its opponent from doing the same. Each agent attempts to choose the best possible action based on the game's state. Similar to Tic-Tac-Toe, the game ends with a draw if both players play optimally.

**Experiment Setup and Results** We conduct experiments with two configurations: (1) Connect-3 on a $3 \times 3$ board and (2) Connect-4 on a $4 \times 4$ board. Similar to the Tic-Tac-Toe game, STRIDE simulates the Breadth-First Minimax algorithm with pruning (see Algorithm 11) to find the optimal action in Connect-N. It first simulates every possible move and scores each node at each game's depth (1 for a win, -1 for a loss, and 0 for a tie or non-leaf node), then uses backward induction to update the scores for each game state. Using its working memory, STRIDE stores the computed scores for all possible actions at various depths. When the game starts, it selects the best action based on the computed scores. The results (averaged over 10 runs) are summarized in Tables 9 and 10.

Table 9: Model performances in Connect-3 (10 runs).

| Outcome | zero-shot CoT | zero-shot CoT w/ code | STRIDE |
|---|---|---|---|
| X Wins (%) | 60 | 90 | 30 |
| Tie (%) | 40 | 0 | **70** |
| O Wins (%) | 0 | 10 | 0 |

We provide the following operational tools to STRIDE to help it emulate Algorithm 11:

- `CalculateScores`: expand every action at each depth and calculate the score for the nodes.
- `GetScores`: retrieve the computed scores for all the actions at the specified depth of the game tree.

## D.2  MDPs WITH LARGER STATE AND ACTIONS SPACES

As discussed in Section 3, since the detailed computations are encapsulated inside the operations, change in the size of state and action spaces does not affect the difficulty of reasoning for problems

Table 10: Model performances in Connect-4 (10 runs).

| Outcome | zero-shot CoT | zero-shot CoT w/ code | STRIDE |
|---|---|---|---|
| X Wins (%) | 50 | 80 | 50 |
| Tie (%) | 10 | 0 | **50** |
| O Wins (%) | 40 | 20 | 0 |

like MDPs. To provide a stronger support of this argument, we conducted additional experiments on the tabular MDP and linear MDP environments, by varying the number of states and actions in the range of (100, 500). The results are reported in Table 11. We can see that the success rate in computing the optimal policy remains relatively the same despite the fact that the size of state and action spaces is increasing.

Table 11: Success rate in taking the optimal action under tabular and linear MDPs.

| Environment | A | S | d | STRIDE |
|---|---|---|---|---|
| Tabular MDP | 50 | 100, 300, 500 | / | 0.99, 0.94, 0.97 |
| | 100 | 100, 300, 500 | / | 0.98, 0.96, 0.96 |
| Linear MDP | 50 | 100, 300, 500 | 5 | 0.96, 0.98, 0.94 |
| | 100 | 100, 300, 500 | 5 | 0.97, 0.96, 0.98 |

