# OpenReview forum: "STRIDE: A Tool-Assisted LLM Agent Framework for Strategic and Interactive Decision-Making"
_ICLR.cc/2025/Conference — Submitted to ICLR 2025_

### Official Review · Reviewer_8RTx · 2024-10-27

**Soundness:** 2
**Presentation:** 2
**Contribution:** 2
**Rating:** 5
**Confidence:** 3

**Summary:**

This paper mainly claims to propose a new framework with specialized tools and memory modules to improve the decision-making capability of LLM-based agents. It analyzes the existing challenges of LLM-based agents. The experimental results show the performance of STRIDE.

**Strengths:**

Pros:
- This paper has analyzed several critical issues of LLM-based agents, which are important to solve.
- This paper provides a new framework specifically for strategic and interactive decision-making.
- The experimental results show the improving performance of STRIDE.

**Weaknesses:**

Cons:
- I think the operation library is very similar to the tool library of tool learning in LLM-based agents, where there have been so many research papers. So what is the significant difference between STRIDE and tool learning methods for LLM-based agents?
- I think a great idea is synthesizing new operations by defining its name and expected functionality. Could you please demonstrate more details above how you create these new operations? I'm still confused about section 3.2.
- I'm also curious about how you implement the working memory, which may utilize retrieval methods or just context windows. The memory mechanism can be greatly important for complex tasks.

**Questions:**

Please see the cons, and please point out if I have any misunderstandings.

---

### Official Review · Reviewer_5q6Q · 2024-10-29

**Soundness:** 1
**Presentation:** 2
**Contribution:** 1
**Rating:** 3
**Confidence:** 4

**Summary:**

The paper presents an agent framework, STRIDE, that uses prebuilt and synthetically generated function calling to improve strategic decision making in LLM agents. The authors test their method on a set of games fulfilling some qualifying criteria and compare the results against other methods towards the same objective.

**Strengths:**

It looks like a lot of thought was put into the presentation of the Algorithms and their outlines in the appendix.

**Weaknesses:**

### Originality
It's not clear to me how this is different from standard function calling / tool use, LLM's generating new operations in a largely robust manner was done with Voyager[0].

[0]: https://arxiv.org/abs/2305.16291

## Quality
The benchmarks used to test STRIDE-SynOp aren't robust in a couple areas:
- the limited results show that SynOp hardly improves performance (and did worse in the second test).
- only 10 runs are used and with higher numbers we would get a better understanding of how well it actually does. (This is not "extensive evaluation")
- If I understand correctly, it's only tested on generating two functions, the original pre-defined functions; there's no information on how it does on any other types of functions.

### Related works:
- The Related Works section "Applications of LLM-based Agents beyond Strategic Reasoning" seems quite unrelated to entire paper.
- Third paragraph is only referencing 4 papers.

## Clarity
Generally, this paper is hard to follow.

Section 4 mentions "To evaluate whether STRIDE can reliably solve new problem instances given provided demonstrations, we repeat experiments on randomly sampled instances and report the averaged results"; it's not clear what the "provided demonstrations" are (though they're mentioned multiple times throughout the paper).

In the appendix, Table 8 and 9 rotate their rows and columns which is confusing, Table 9 I'm not sure if the 90 of `zero-shot CoT w/ code` should be the best result, this is hard to follow.

The paper is more concerned with "fiduciary agents" than "strategic reasoning" yet strategic reasoning is nominal focus of the paper. The paper starts to shift more towards this aim throughout the paper, which isn't made clear.


### Less important but should be addressed:
- Numerous areas have a lack of consistent formatting: *thought* sequence is sometimes not italicized, GPT-3.5-Turbo-0125 is sometimes with / without code styling
- Spelling errors
	- "through a sequence of structured Thought unit"  -- 'units'
	- "a set pf Python functions taking care"
	- "on MDP under both known model"

**Questions:**

- STRIDE-SynOp shows minimal improvement over few-shot CoT. Could you explain the cost-benefit analysis of using STRIDE-SynOp versus simpler approaches?
- Most of the experiments only use 10 runs, why was this?

---

### Official Review · Reviewer_5yf9 · 2024-11-01

**Soundness:** 3
**Presentation:** 2
**Contribution:** 2
**Rating:** 5
**Confidence:** 3

**Summary:**

The paper proposes a framework designed to address specific challenges associated with using large language models in strategic environments. The authors identify several limitations of LLMs, including:
(1) difficulty in interpreting game rules,
(2) challenges in long-horizon planning,
(3) poor performance in unknown environments, and
(4) limitations in scenarios involving multiple agents or opponents.

The proposed framework addresses these issues by processing inputs through a "thought unit" that generates (1) summarized information, (2) a list of operations to execute, and (3) an exit indicator.
To prevent the need for LLMs to perform lower-level computations or generate potentially error code, the framework prepares an operation database filled with predefined functions. Additionally, this function database can be automatically updated as needed by the thought unit. The framework includes a working memory component designed to retain important information for managing long-term context.
The authors apply this framework to several scenarios where the challenges are most significant, comparing its performance against benchmarks and demonstrating descent improvements.

**Strengths:**

The STRIDE framework addresses LLM challenges by effectively decomposing complex strategic tasks into several components. It leverages Large Language Models for high-level reasoning and decision-making, while delegating computation and execution to a well-structured operation database.
This separation coulud efficiency allow the LLM to focus on strategic aspects without low-level execution details.
Additionally, the framework is supported by components for error handling and working memory, ensuring smooth and reliable processing.

**Weaknesses:**

Dependency on Predefined Operations:
STRIDE heavily relies on a predefined set of operations. It is benefit to avoid LLM to do the low-level calculation, However, this reliance introduces challenges related to the selection and management of these operations. Specifically, determining which operations to activate in complex scenarios can be challenge. As the function database expands, ensuring that operations are distinct and appropriately invoked becomes increasingly difficult.
Additionally, within the thought unit, there may be proposals for new functions that replicate or overlap with combinations of existing operations. If these new functions are added without rigorous validation, they might replace existing ones, even if they are less effective or appropriate.
To mitigate this risk, integrating a filtering mechanism that assesses the validity and necessity of new functions before they are added to the database could be useful. This mechanism would help maintain the integrity and efficiency of the operation database, ensuring that only beneficial updates are implemented.

**Questions:**

(1) For the first challenges authors tried to address, (i) LLM may fail to accurately interpret game rules and objectives expressed in natural language. Then authors claim, this challenge can be addressed by executing an operation that evaluates the agent's utility in the Thought unit.
In your reference in the appendix, your prompt of input are the detailed description of the game rules.
There could be a gap between interpreting game rules and calling the operation to calculate utility. The framework still heavily relies on accurately understanding the comprehensive description of the game rules. LLM still needs to read the entire document to determine when to invoke the function and how to assess utility. The current method of calling the operation primarily ensures the accuracy of the computation rather than using LLM' ability to calculate.

(2) In Tables 3 and 5, the few-shot CoT with code performs worse than the zero-shot CoT, whereas in Table 1, the few-shot CoT with code significantly outperforms the zero-shot CoT. Could you explain the potential reasons for these inconsistence across different tasks?

(3) In the framework, suppose the function database initially contains no operations and relies solely on the error handling component to design functions and add them. Would this approach be equivalent to a zero-shot CoT with code? Considering the performance of zero-shot CoT with code in Tables 3 and 5, this situation underscores the point mentioned in the weakness part about the need for a filtering mechanism to fillter functions before they are added to the database.

---

### Official Review · Reviewer_ivRA · 2024-11-04

**Soundness:** 3
**Presentation:** 4
**Contribution:** 3
**Rating:** 6
**Confidence:** 3

**Summary:**

The paper introduces STRIDE, a novel framework designed to enhance the strategic and interactive decision-making capabilities of large language models (LLMs). Recognizing the limitations of current LLMs in strategic multi-agent environments—such as poor mathematical reasoning, difficulty in following instructions, and a tendency to generate incorrect information—the authors propose equipping LLMs with external memory and specialized tools. STRIDE operates by orchestrating a sequence of structured Thought units, each containing operations that execute predefined Python functions for low-level calculations and reasoning tasks. The framework is evaluated across several economically significant environments, including bilateral bargaining games, Markov Decision Processes (MDPs), and dynamic mechanism design problems. Experimental results demonstrate that STRIDE significantly outperforms baseline methods, enhancing LLMs' strategic reasoning and decision-making abilities.

**Strengths:**

1. Originality: The paper introduces an innovative framework that enhances LLMs with external tools and memory, addressing specific limitations in strategic reasoning tasks. This integration allows LLMs to perform complex computations and maintain important parameters throughout interactions.
2. Quality: The experimental evaluation is thorough, covering multiple strategic decision-making problems. The use of quantitative metrics provides strong evidence for the improvements offered by STRIDE over baseline methods.
3. Clarity: The paper is well-structured, with clear explanations of the challenges faced by LLMs and how STRIDE addresses them. Figures and examples, such as the bargaining game illustration, effectively aid understanding.
4. Significance: Enhancing LLMs for strategic and interactive decision-making has important implications for deploying AI agents in real-world competitive environments. The work advances the field by providing a framework that can be extended to various complex tasks.

**Weaknesses:**

1. While the integration of external tools and memory is effective, similar approaches have been explored in previous works. The paper could better highlight how STRIDE distinguishes itself from existing frameworks.
2. The experiments focus on specific problem instances. Expanding the evaluation to a broader range of strategic environments would strengthen the claims about STRIDE's generalizability. The scalability analysis focuses mainly on state/action space size rather than more complex strategic scenarios

**Questions:**

1. How does STRIDE perform in other strategic environments not tested in the paper, such as more complex games or real-world economic scenarios? Can the framework easily adapt to these new settings?
2. Can you provide a more detailed comparison between STRIDE and other LLM agent frameworks that utilize tools and memory? What are the unique advantages of STRIDE?
3. Scalability and Computational Efficiency: How does the framework handle scalability concerning the size of the operation library and the complexity of tasks? Were there any computational constraints observed during the experiments?
4. Operation Synthesis Limitations: When STRIDE synthesizes new operations, are there limitations in terms of reliability and correctness of the generated code? How does this affect the overall robustness of the framework?

---

### Meta-Review · Area_Chair_PmcZ · 2024-12-07

**Metareview:**

The paper proposes the STRIDE framework for addressing LLMs' weaknesses at strategic decision-making. STRIDE main technical approach is equipping LLMs with memory and a specialized toolkit for addressing the LLMs' decision-making weaknesses.

STRIDE's strength is the lift in performance it provides on some of the (highly nontrivial) tasks that serve as the paper's benchmarks. The weaknesses are its similarity to existing frameworks for addressing the same problem, the reliance on LLMs' detailed understanding of task descriptions, which is a major factor in LLMs' decision-making failures, and the reliance on a predefined operator toolkit.

The single most promising way to improve the submission would be to articulate better, in a revised submission version and/or in the rebuttals, STRIDE's novelty w.r.t. related work.

However, the authors haven't submitted any rebuttals and, with the aforementioned weaknesses unaddressed, this submission isn't ready for publication.

**Additional Comments On Reviewer Discussion:**

The authors haven't submitted any rebuttals and there has been no discussion, since all but one reviewer put the paper below the acceptance threshold to begin with.

The metareviewer finds the weaknesses identified by the reviewers substantial and therefore recommends rejection.

---

### Decision · Program_Chairs · 2025-01-22

Reject